# SQ-FORMAT: A UNIFIED SPARSE-QUANTIZED HARDWARE-FRIENDLY DATA FORMAT FOR LARGE LANGUAGE MODELS

## ABSTRACT

Post-training quantization (PTQ) plays a crucial role in the democratization of large language models (LLMs). However, existing low-bit quantizaiton and sparsification techniques are difficult to balance accuracy and efficiency due to the limited hardware support. For example, W4A8 can only achieve the same peak TOPS as W8A8 whereas the GPU-supported sparse data format (2:4 semi-structure sparse) is seldomly adopted due to the loss of accuracy. To bridge this gap, in this paper, we propose the **Sparse-Quantized Format** (SQ-format), which is a unified data format for quantization and sparsification potentially easily supported by new hardware and existing GPUs. SQ-format makes use of the fact that sparse matrix can be accelerated in high-precision, and low-precision matrix multiplication can also be accelerated accordingly. As such, SQ-format is proposed to achieve Pareto improvement between performance and throughput. This format is particularly suitable for activations with outlier inequality status and makes their static compression possible. We show the state-of-the-art PTQ performance with SQ-format, propose the hardware required to support it, and further offer the design exploration and insights for the next-generation AI acceleractors.

## 1 INTRODUCTION

Large Language Models (LLMs) have demonstrated remarkable capabilities (Wei et al., 2025; Li et al., 2025), but their substantial memory and computation requirements present significant deployment challenges (Chavan et al., 2024). To address this, quantization (Xiao et al., 2023; Lin et al., 2024; Liu et al., 2025) and sparsification (Frantar & Alistarh, 2023; Dettmers et al., 2024) have emerged as indispensable techniques, enabling LLMs to run on fewer GPUs or more cost-effective hardware. Post-Training Quantization (PTQ) is particularly appealing as it allows for the direct quantization of pre-trained models without costly retraining.

Current PTQ practices (Lin et al., 2023; Frantar et al., 2022a) have established that an 8-bit weight, 8-bit activation (W8A8) format can maintain near-lossless accuracy compared to BF16 setting. The primary challenge now lies in pushing beyond this frontier to even lower bit-widths. However, advanced uniform quantization, such as a W4A4 format, often leads to a significant degradation in model performance (Liu et al., 2025). Furthermore, a critical hardware-algorithm gap impedes the practical realization of mixed-precision benefits. Although modern GPUs feature specialized tensor cores for 4-bit and 8-bit computations, they typically lack native support for direct hybrid-precision operations (e.g., INT4 matrix multiplied by INT8 matrix). Consequently, theoretically efficient schemes like W4A8 must be emulated using higher-precision data paths, nullifying the potential performance gains, creating a persistent gap between theoretical throughput and deployed reality (Lin et al., 2024; Zheng et al., 2024).

The solution lies in designing a hardware-friendly data format natively supporting sparsification and quantization in hybrid precisions to accelerate computation while maintaining the performance. To maintain the model performance, Xiao et al. (2023); Lee et al. (2024) have revealed that a small subset of values in both weights and activations disproportionately impacts model performance and is highly sensitive to quantization errors. Whether through quantization or sparsification, existing compression methods struggle to adapt to the non-uniform distribution of information in LLMs.

Quantization formats like W$x$A$y$ apply a uniform bit-width across entire tensors, while sparsification approaches like NVIDIA 2:4 semi-structure sparse (Pool et al., 2021; Frantar & Alistarh, 2023) lacks the flexibility to handle non-uniform information (Figure 6). These observations motivate a shift towards a fine-grained, unified sparse and quantized paradigm, where critical values are preserved in higher precision while the majority are compressed to lower bit-widths (e.g., 4-bit or even 2-bit). Consider a W4A8 setting: current hardware falls back to the less efficient W8A8 computation path. Our core idea is that if the 8-bit activations could be decomposed into a sparse, high-precision component (8-bit) and a dense, low-precision component (4-bit) in a hardware-friendly way (Zhong et al., 2024a), the computation burden can be shifted to the much faster 4-bit tasks.

In this paper, inspired by the co-design philosophy evident in NVIDIA's NVFP4 (Alvarez et al., 2025) and 2:4 semi-structure sparse formats (Pool et al., 2021), where hardware features and data formats are developed in tandem, we propose **Sparse-Quantized Format** (SQ-format). This format enables hybrid-precision compression frameworks. In summary, our contribution are as follows:

- **SQ-format Definition and Its Hardware Implementation (Section 2).** We define SQ-format, a new data format designed for hybrid integer precision matrix multiplication. We provide its feasible hardware implementations for both current GPUs and next-generation AI accelerators.
- **Achieving Pareto Improvement between Accuracy & Throughput (Section 3.2).** We develop two algorithms leveraging SQ-format for weights and activations. Conducting extensive experiments on various LLMs, our results demonstrate that SQ-format elevates schemes with W4A4-level throughput to near W4A8-level accuracy. Compared to W4A8 baselines, SQ-format delivers superior throughput with a model performance gap of less than 1%.
- **Quantization for Static Activation Splitting (Section 3.3).** Applying SQ-format to activations introduces potential runtime overhead due to dynamic value selection is not natively well supported by GPUs. To further eliminate this bottleneck, we develop a static strategy that pre-determines the high-precision components by analyzing activation-weight product contributions on a calibration set. This strategy removes the costly runtime operation, making the solution more hardware-friendly while achieving performance comparable to its dynamic counterpart.

Through extensive experiments, this paper validates the superiority of SQ-format. We contend that our work not only offers a practical solution for accelerating LLMs on current hardware but also presents a compelling co-design blueprint for future AI accelerators.

## 2 METHODOLOGY

In this section, we first define the **Sparse-Quantized Format** (SQ-format) in Section 2.1. Then, we demonstrate two example algorithms leveraging SQ-format for weights and activations in Section 2.2. We finally discuss the hardware design solutions for SQ-format in Section 2.3.

### 2.1 SQ-FORMAT DEFINITION

SQ-format only applies to one operand in matrix multiplication to avoid complicated cross-format computation. The other operand can use normal uniform formats. SQ-format supports both integer (e.g., INT4) and floating point (e.g., FP4). It divides the precision used into high-precision $h_{\text{high}}$ and low-precision $h_{\text{low}}$ (e.g., $h_{\text{high}} = \text{INT8}, h_{\text{low}} = \text{INT4}$). The matrices in SQ-format are divided by bank. A fixed bank size $b$ and sparsity $s$ avoids the load imbalance problem and removes the distributed accumulator problem encountered with unstructured sparsity, and further reduce the energy- and area-hungry (Liu et al., 2021; Moor Insights & Strategy, 2020). This is the reason that GPU adopts 2:4 semi-structure sparse and our proposed SQ-format can be viewed as a general extension, where $s = 0.5, h_{\text{low}} = 0, b = 4$. We define the SQ-format components in Equation (1).

$$\text{SQ-format}(\mathbf{X}) = ([\mathbf{X}_{\text{quant}}], [\mathbf{S}_{\text{quant}}], [\mathbf{m}], h_{\text{high}}, h_{\text{low}}, b, s) \tag{1}$$

With $\mathbf{X}$ as a general matrix, $[\mathbf{X}_{\text{quant}}], [\mathbf{S}_{\text{quant}}]$ are the quantized matrix with $h_{\text{high}}/h_{\text{low}}$ and the scaling matrix. We determine high- and low-precision parts by masks $[\mathbf{m}]$, which can be represented implicitly by low-precision parts or explicitly by additional vectors. This serves as a framework whose concrete mathematical formulation varies with the hardware implementation.

## 2.2 EXAMPLE ALGORITHMS WITH SQ-FORMAT

We demonstrate two PTQ algorithms based on SQ-format. For weights and activations, each algorithm selects one to use SQ-format, and the other is executed with classic quantization.

**SQ-format on Weights.** When using SQ-format on weights, a weight matrix is represented by high-precision and low-precision parts. For the simplicty of hardware implementation, we adopt the symmetric quatization. Therefore, the unused largest value of the low-precision format is considered as a high-precision mask, indicating that high-precision elements should be used for computation here. For example, When $h_{\text{low}} = $ INT2, values in $\{-1, 0, 1\}$ are normal elements, and $v_{\text{mask}} = 2$ is used to indicate the high-precision presence. The high-precision elements are sparsely located, while stored compactly since we adopt a fixed-sparsity per-bank setting without any wasted gaps. See an INT4/INT8 example in Figure 1.

**Algorithm 1 (SQ-format on Weights)**: Based on SQ-format, we combine GPTQ (Frantar et al., 2022a) and SmoothQuant (Xiao et al., 2023) to implement our quantization strategy. We first smooth the weight matrices on the calibration set. When quantizing the smoothed weight matrix $\mathbf{W}'$ of a given layer, we use the Hessian matrix $\mathbf{H}$, derived from the calibration set activations, to approximate information loss. To determine the high- and low-precision elements, follow Frantar et al. (2022a;b), for $\mathbf{W}'_{r,i}$, we compute its importance score by $I_{r,i} = (\mathbf{W}'_{r,i})^2 / (\mathbf{H}_{i,i}^{-1})^2$, which synthesizes the weight's own magnitude with the model's sensitivity to its perturbation. Within each weight bank, we rank the weights based on $I$ and select the top $(1-s)$ fraction of weights within each column to generate a high-precision mask. The detailed procedure is described in Algorithm 1.

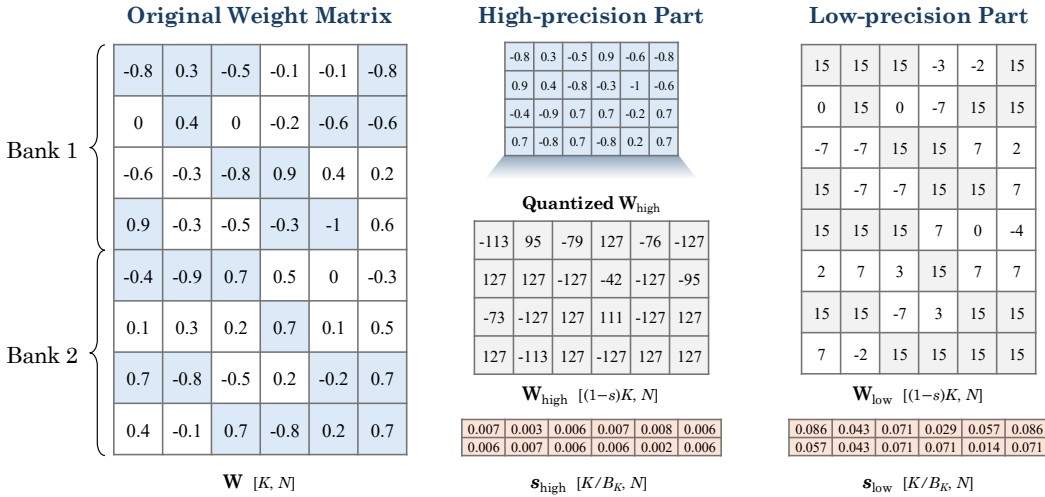

Figure 1: An example of a weight matrix using SQ-format ($h_{\text{high}} = $ INT8, $h_{\text{low}} = $ INT4, $s = 0.5$). In this case, each column of each bank is split into a high-precision part and a low-precision part for quantization as a group. The high-precision part is stored compactly, while the low-precision part is still stored in its original shape, but the corresponding high-precision places is valued $v_{\text{mask}} = 15$.

**SQ-format on Activations.** When using SQ-format on activations, it divides matrices into two parts: high-precision (important) and low-precision (less important), whereas there are naturally per-channel outliers in activations, which are of significant importance to information loss. However, we cannot quantize activations like Algorithm 1 owing to their dynamic nature. During inference, we need to construct SQ-format representation for activations, which introduces extra TopK overhead in addition to (de)quantization. The overhead can be easily eliminated with dedicated hardware support illustrated in Section 2.3. SQ-format is bank based and thereby can be mapped on the hardware implementation as a pipelined preprocess for tensor cores.

**Algorithm 2 (Static Strategy for SQ-format on Activations)**: To broaden the SQ-format application, we also propose a static strategy for activations to address this issue. We can generate mask vectors for SQ-format on the calibration set in advance to select high- and low-precision elements by channel. During inference, we separate the computation tasks to two parts by masks. The lower-

precision part brings extra throughput benefits. In early experiments, we find that generating static activation masks based solely on the absolute values of activations ($I_j = |\bar{\mathbf{A}}_j|, j \in [1, K]$) leads to significant performance degradation. This is reasonable since we cannot approximate the dot product $\mathbf{A} \cdot \mathbf{W}$ by activations only. To this end, we redefine the importance score by considering the per-channel average magnitude of $\mathbf{A} \cdot \mathbf{W}$.

Specifically, we first collect per-channel average activations $\bar{\mathbf{A}}_j$ for each input channel $j$ over a calibration set. Given a smoothed weight matrix $\mathbf{W}'$, the importance score $I_j$ for channel $j$ is $I_j = |\bar{\mathbf{A}}_j \cdot \sum_i \mathbf{W}'_{i,j}|$. We then generate mask according to $I$, selecting the top $(1 - s)$ fraction channels within each bank. Furthermore, as part of the PTQ process, the columns of the weight matrices can be reordered according to $I$. This alignment can improve data locality for hardware kernels. The detailed procedure is described in Algorithm 2.

The additional storage overhead for static mask is negligible due to its per-channel nature. For instance, the mask size for Llama-3-70B is only 5.94 MB.

---

**Algorithm 1** SQ-format on Weights

1: **Input:** Weight $\mathbf{W} \in \mathbb{R}^{K \times N}$, calibration dataset $\mathcal{D}$, sparsity $s$, bank size $b$, high-precision $h_{\text{high}}$, low-precision $h_{\text{low}}$.

2: $\mathbf{W}', \mathbf{H} \leftarrow \text{Smooth}(\mathbf{W}, \mathcal{D})$
3: $\mathbf{I} \leftarrow (\mathbf{W}')^2 / (\text{diag}(\mathbf{H}^{-1}))^2$

4: **for** each bank $\mathbf{w}$ of $\mathbf{W}'$ **do**
5:     Generate Precision Mask $\mathbf{m_w}$
            ▷ Select top $(1 - s)$ elements by $\mathbf{I_w}$
6:     $(\mathbf{w}'_{\text{high}}, \mathbf{s}'_{\text{high}}) \leftarrow \text{Quant}(\mathbf{w}' \odot \mathbf{m_w})$
7:     $(\mathbf{w}'_{\text{low}}, \mathbf{s}'_{\text{low}}) \leftarrow \text{Quant}(\mathbf{w}' \odot \sim \mathbf{m_w})$
8: **end for**

9: Generate $(\mathbf{W}_{\text{high}}, \mathbf{W}_{\text{low}}, \mathbf{S}_{\text{high}}, \mathbf{S}_{\text{low}})$.

**Algorithm 2** SQ-format on Activations (Static)

1: **Input:** Weight $\mathbf{W} \in \mathbb{R}^{K \times N}$, calibration dataset $\mathcal{D}$, sparsity $s$, bank size $b$, high-precision $h_{\text{high}}$, low-precision $h_{\text{low}}$.

2: $\mathbf{W}', \bar{\mathbf{A}} \leftarrow \text{Smooth}(\mathbf{W}, \mathcal{D})$

3: **for** each bank $\mathbf{w}$ of $\mathbf{W}'$, $\bar{\mathbf{a}}$ of $\bar{\mathbf{A}}$ **do**
4:     $I_j \leftarrow |\bar{\mathbf{A}}_j \cdot \sum_i \mathbf{W}'_{j,i}|, \forall j \in \text{bank}$
5:     Generate Precision Mask $\mathbf{m_w}$
            ▷ Select top $(1 - s)$ elements by $I_j$
6:     $(\mathbf{w}', \mathbf{s}') \leftarrow \text{Quant}(\mathbf{w}, h_{\text{high}}, h_{\text{low}})$
7: **end for**
8: Reorder rows of $\mathbf{W}'$ based on mask $\mathbf{m}$.

9: Generate $(\mathbf{W}_{\text{quant}}, \mathbf{S}_{\text{quant}}, \mathbf{m})$.

## 2.3 HARDWARE DESIGN

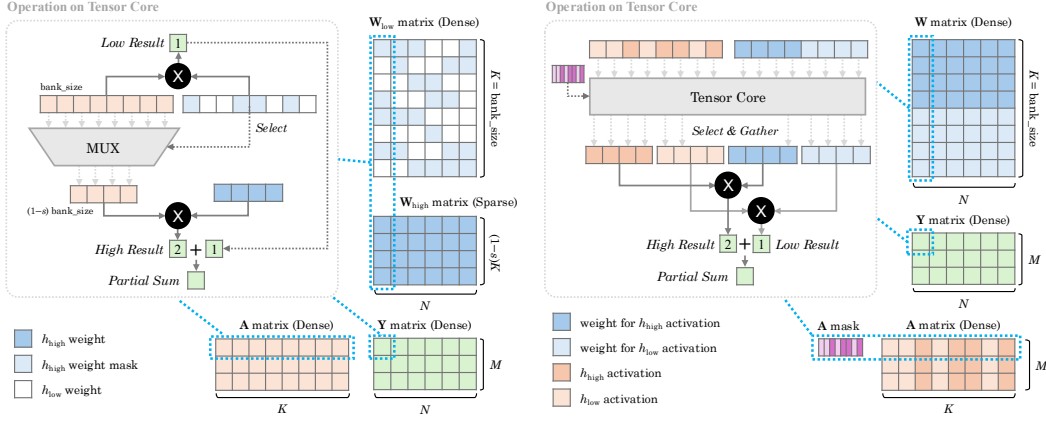

(a) SQ-format for weights.          (b) Static SQ-format for activations.

Figure 2: Hardware implementation for applying SQ-format on weights and activations.

Native support for SQ-format requires dedicated hardware accelerators to fully leverage its advantages in improving accuracy and throughput. The core challenges are: (1) when applying SQ-format on weights, we need to process the compactly stored high-precision parts efficiently; and (2) when using dynamic SQ-format on activations, we need a pipelined unit to dynamically compute the high-

precision masks. These hardware units can function as dedicated accelerators (e.g. gather, scatter, etc) within AI chips, working in conjunction with standard tensor cores.

As illustrated in Figure 2a, we present the computation path for applying SQ-format on weights with dedicated hardware support, where high and low-precision computations are split into two parallel paths. The low-precision part can be directly processed by tensor cores with dense matrix multiplication. The high-precision part, however, requires detecting the $v_{mask}$ values in the low-precision part to gather corresponding elements from the activation tiles before sending them to tensor cores. The advantage is that since the high-precision part can be over 8x sparse, the latency of this parallel path will be completely hidden by the low-precision computation.

As illustrated in Figure 2b, we demonstrate the computation path for applying static SQ-format on activations on GPUs. It relies on pre-computed activation masks to divide the entire computation into two paths with different precisions. Since the PTQ process has already reordered the banks of weight matrices, we only need selection and gathering operations for the activation tiles. Although the high- and low-precision computation streams are executed serially, the overall throughput increases because a significant portion of the computation (can up to 3/4) is converted to low-precision. We implemented the kernel for computing static SQ-format on activations with CUTLASS library on GPUs to demonstrate its practicality. As shown in Figure 3, our implementation achieves higher throughput than W8A8. A larger sparsity leads to greater throughput, bringing the performance progressively closer to the theoretical ceiling of W4A4.

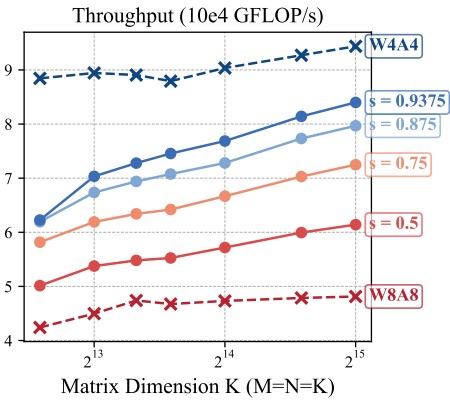

Figure 3: Throughput of static SQ-format on activations from $s = 0.5$ (2x sparse) to $0.9375$ (16x sparse).

Dynamic strategy for SQ-format on activations requires additional hardware support, as shown in Figure 4. When implemented on GPUs, we need to use CUDA cores to compute the high-precision mask for the entire activation tensor. However, an additional pipelined hardware unit could ensure that the mask for each bank is computed and ready before tensor cores execute it, thereby achieving throughput performance comparable to the static strategy.

To validate the hardware feasibility, we implement the proposed unit in RTL and synthesized it using the TSMC 12nm process library. We compare it against a standard integer MAC array under 12-bit I/O. Even accounting for the additional gather unit requires for dynamic masking, the overall silicon area is reduced by 35.8% compared to the baseline. Detailed synthesis results are provided in Appendix C.

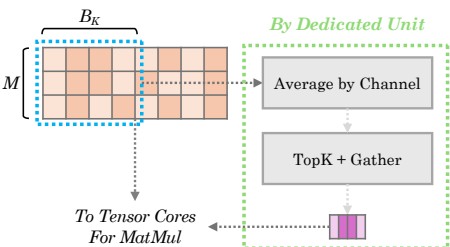

Figure 4: Dedicated hardware supporting dynamic activation SQ-format.

## 3 EXPERIMENTS & FINDINGS

To validate the effectiveness of SQ-format and to investigate why it is effective, we conduct extensive experiments on different LLMs, including dense models and MoE models.

### 3.1 EXPERIMENTAL SETUPS

**Models and Benchmarks.** Our quantized models include: Llama-3-8B, Llama-3-70B (Team, 2024), Qwen3-30B-A3B (Team, 2025). We compare normalized accuracy on six non-generative datasets: ARC-easy, ARC-challenge (Clark et al., 2018), OpenBookQA (Mihaylov et al., 2018), Hellaswag (Zellers et al., 2019), Winogrande (Sakaguchi et al., 2020), and PIQA (Bisk et al.,

2020); two generative datasets: GSM8k (Cobbe et al., 2021) and AGIEval (Zhong et al., 2024b), as well as perplexity on Wikitext (Merity et al., 2017) and Lambada (Paperno et al., 2016) with lm-evaluation (Gao et al., 2024). GSM8k is evaluated in 8-shot setting while others are zero-shot.

**Notation.** We conduct the experiments with integer format to validate the effectiveness. The configuration of SQ-format is denoted as $B - (h_{\text{high}}/h_{\text{low}}) - s$, where $B$ is the bank size. The configuration based on Algorithm 1 is denoted as W(SQ$x$)A$y$, where W uses SQ-format with an equivalent of $x$ bits and A uses INT$y$ quantization. The equivalent bit number is $x = (1-s)h_{\text{high}} + h_{\text{low}}$. Similarly, the configuration based on SQ-format on activations (dynamic and static strategy based on Algorithm 2) is denoted as W$x$A(SQ$y$). The equivalent bit number is $y = (1-s)h_{\text{high}} + sh_{\text{low}}$. The static strategy also includes an additional 1-bit per-channel mask, while it is negligible for $y$.

**Baselines.** As a unified sparse-quantized data format, we have chosen post-training sparse baselines: SpQR (Dettmers et al., 2024) and SparseGPT (Frantar & Alistarh, 2023). For PTQ algorithms, we compare with two accuracy levels of W4A8 and W4A4, and compare with three baselines: GPTQ (Frantar et al., 2022a), SmoothQuant (Xiao et al., 2023), and SpinQuant (Liu et al., 2025).

For all experiments, our calibration set consists of 32 randomly sampled text segments from Wikitext, each with a length of 2048 tokens. We present $B - (8/4) - s$ implementation that runs on modern GPUs. In Section 4 and Appendix A.1, we provide more results including $(8/3), (8/2), (4/2)$ settings. We conducted experiments on different bank sizes (4, 8, 16, 32, 64 and 128) and sparsity (0.5, 0.75, 0.875, and 0.9375) grids. We select the best from the grid search results, while complete results will be provided in Appendix A.1. To demonstrate the applicability of SQ-format on other data types, we also show the results of applying SQ-format on DeepSeek-R1 weights with FP8/FP4 quantization and 4x sparsity in Appendix B.

## 3.2 FINDING 1: SQ-FORMAT ACHIEVES ACCURACY-THROUGHPUT PARETO IMPROVEMENT

We obtain insights about the Pareto improvement between accuracy and throughput caused by the SQ-format from two sets of comparisons. The benchmarking results are shown in Table 1.

**W4A8 vs. W4A(SQ).** GPUs cannot efficiently compute W4A8 algorithms; in fact, it will fallback to W8A8 execution path. However, SQ-format separates A4 and A8 computations, transforming most of the original computing tasks into W4A4 computations, significantly improving throughput. For performance, W4A(SQ6) and W4A(SQ5) achieve average accuracy and perplexity on par with GPTQ on Llama-3 models, while on Qwen-3, the average benchmark accuracy improves by 3.87% and achieves a better perplexity.

We illustrate the Pareto frontier in Figure 5. The x-axis represents the speedup derived from end-to-end prefilling latency measurements using SQ-format with the static activation strategy. As shown, SQ-format effectively bridges the gap between the high-precision W4A8 and the high-efficiency W4A4. Notably, for larger models like Llama-3-70B, SQ-format achieves a speedup of $1.71\times$, capturing $\sim$89% of the theoretical W4A4 acceleration while maintaining higher model performance. Detailed latency and bandwidth results are provided in Appendix D.

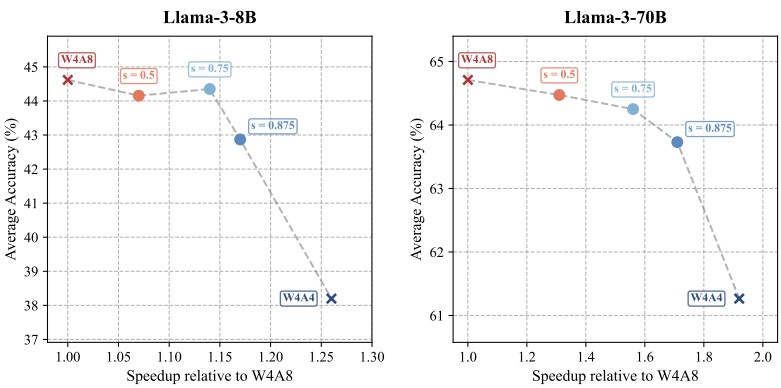

Figure 5: Accuracy-Speed Pareto frontier on Llama-3 models.

Table 1: Performance comparison of quantization methods. For each comparison set, **Red bold values** highlight the best performance, and **Black bold values** highlight the second best performance. In this table, W$x$A(SQ$y$) results are using dynamic settings simulated on GPUs.

| Model | Method | Setting | Non-generative Acc (%) | | | | | | Generative Acc (%) | | Perplexity | |
|---|---|---|---|---|---|---|---|---|---|---|---|---|
| | | | ARC-c (↑) | ARC-e (↑) | PIQA (↑) | Hellas. (↑) | Wino. (↑) | OBQA (↑) | GSM8k (↑) | AGIEval (↑) | Wiki. (↓) | Lamb. (↓) |
| Llama-3 8B | BF16 | W16A16 | 53.41 | 77.74 | 80.85 | 79.21 | 73.40 | 45.00 | 49.43 | 33.68 | 7.26 | 3.09 |
| | SpQR | W4A16 | 52.13 | 76.81 | 79.87 | 78.84 | 73.40 | 45.20 | 45.79 | 32.78 | 7.55 | 3.22 |
| | SparseGPT | W16A16 | 31.74 | 58.96 | 68.88 | 53.69 | 63.38 | 33.80 | 2.05 | 25.60 | 18.25 | 9.28 |
| | GPTQ | W4A8 | **53.15** | **78.32** | **79.86** | 77.62 | 73.79 | 44.40 | 40.86 | **32.88** | 7.98 | **3.41** |
| | SmoothQuant | W4A8 | 37.20 | 59.97 | 72.63 | 67.49 | 67.56 | 37.40 | 5.07 | 25.71 | 16.51 | 5.87 |
| | SpinQuant | W4A8 | 51.78 | 75.08 | 80.20 | **77.71** | **74.19** | 44.40 | **43.06** | 32.52 | **7.81** | 3.38 |
| | SQ-format | W4A(SQ6) | 52.13 | 77.44 | **79.87** | **78.05** | 73.48 | **45.20** | 40.26 | 32.35 | **7.97** | 3.47 |
| | | W4A(SQ5) | 51.96 | 77.15 | 79.43 | 77.68 | **74.27** | 44.40 | 40.94 | 32.35 | 8.02 | 3.50 |
| | GPTQ | W4A4 | 46.92 | 71.50 | 77.63 | 75.24 | **71.11** | 43.00 | 28.58 | 30.46 | 9.13 | 4.27 |
| | SmoothQuant | W4A4 | 24.23 | 26.80 | 51.90 | 27.08 | 50.90 | 25.00 | 0.00 | 24.72 | >100 | >100 |
| | SpinQuant | W4A4 | **47.95** | **74.66** | 77.53 | 75.28 | 69.53 | 41.60 | 32.90 | **31.14** | 9.08 | 4.11 |
| | SQ-format | W(SQ6)A4 | **49.40** | **75.42** | 79.38 | **77.75** | **71.74** | 44.00 | **39.58** | **31.91** | **8.01** | **3.42** |
| | | W(SQ5)A4 | 47.44 | 73.15 | 76.71 | **76.03** | 69.06 | 42.20 | 33.36 | 30.36 | **8.85** | 3.91 |
| | | W(SQ4.5)A4 | 47.61 | 72.01 | **78.24** | 75.39 | 69.22 | **44.20** | 34.12 | 30.61 | 9.00 | 3.97 |
| Llama-3 70B | BF16 | W16A16 | 64.51 | 86.11 | 84.39 | 84.96 | 80.19 | 48.40 | 81.05 | 45.67 | 2.92 | 2.58 |
| | SpQR | W4A16 | 63.99 | 85.19 | 84.39 | 85.05 | 79.87 | 48.80 | 79.38 | 44.63 | 3.20 | 2.65 |
| | SparseGPT | W16A16 | 48.72 | 74.45 | 77.64 | 70.69 | 74.82 | 41.00 | 31.31 | 29.91 | 9.86 | 3.07 |
| | GPTQ | W4A8 | **62.54** | 84.13 | 83.56 | **84.32** | 80.42 | **48.20** | 78.31 | **45.00** | 3.48 | **2.61** |
| | SmoothQuant | W4A8 | 24.40 | 42.42 | 64.36 | 64.86 | 57.06 | 33.40 | 0.75 | 29.30 | 8.41 | 23.83 |
| | SpinQuant | W4A8 | 62.03 | **86.32** | **84.66** | 84.25 | **81.06** | **49.00** | **79.38** | 42.16 | 3.73 | 2.72 |
| | SQ-format | W4A(SQ6) | **62.80** | **84.81** | **84.06** | **84.44** | 80.19 | 48.00 | 78.32 | **44.68** | **3.31** | 2.65 |
| | | W4A(SQ5) | 61.95 | 84.51 | 83.73 | 84.14 | 80.90 | 47.80 | **79.23** | 44.67 | **3.45** | 2.67 |
| | GPTQ | W4A4 | 59.89 | 81.64 | **83.07** | 83.11 | 77.50 | 46.40 | 74.82 | 40.60 | 4.58 | 2.98 |
| | SmoothQuant | W4A4 | 25.21 | 25.00 | 50.21 | 26.43 | 51.30 | 26.30 | 0.00 | 24.42 | >100 | >100 |
| | SpinQuant | W4A4 | 37.97 | 62.46 | 73.67 | 65.62 | 61.64 | 41.40 | 1.97 | 26.27 | 15.26 | 15.07 |
| | SQ-format | W(SQ6)A4 | **61.01** | **83.12** | 83.03 | **84.10** | 78.77 | 46.40 | **78.62** | **41.84** | **3.79** | **2.66** |
| | | W(SQ5)A4 | **59.98** | **83.08** | 81.88 | **83.15** | **77.82** | **49.40** | 75.82 | 41.47 | 4.49 | 2.69 |
| | | W(SQ4.5)A4 | 59.64 | 81.99 | 82.75 | 83.08 | 77.43 | 47.40 | 75.59 | 39.89 | 4.67 | 2.73 |
| Qwen-3 30B-A3B | BF16 | W16A16 | 55.80 | 79.17 | 80.58 | 77.65 | 70.88 | 45.80 | 90.45 | 63.05 | 10.89 | 4.12 |
| | GPTQ | W4A8 | 52.73 | 77.73 | 79.37 | **76.75** | 68.66 | 44.20 | 82.69 | 60.20 | 11.21 | **4.49** |
| | SmoothQuant | W4A8 | 40.35 | 57.95 | 71.98 | 58.91 | 60.22 | 38.00 | 59.21 | 35.82 | 26.42 | 27.19 |
| | SQ-format | W4A(SQ6) | 55.38 | **79.21** | 79.54 | 76.60 | **70.56** | **44.80** | 88.78 | **61.33** | 11.11 | 4.47 |
| | | W4A(SQ5) | **55.46** | 78.75 | **80.25** | 76.28 | 70.24 | 44.20 | **89.84** | 61.31 | **11.10** | 4.51 |
| | GPTQ | W4A4 | 49.48 | 75.88 | **78.50** | 74.77 | 67.71 | 43.20 | 65.39 | 52.85 | 11.81 | 5.06 |
| | SmoothQuant | W4A4 | 23.20 | 29.29 | 53.42 | 28.35 | 51.46 | 28.20 | 0.00 | 25.95 | >100 | >100 |
| | SQ-format | W(SQ6)A4 | **53.75** | **76.22** | 78.02 | **75.58** | **70.64** | **44.40** | **88.17** | **56.57** | **11.43** | **4.86** |

**W4A4 vs. W(SQ)A4.** By introducing sparse high-precision elements, the accuracy of W4A4 quantization can be economically improved. From 16x to 4x sparse, the average accuracy improvement ranges from 0.11% to 3.54%. Among them, 4x sparse setting W(SQ6)A4 achieves accuracy close to W4A8. It also allows true benefits to be realized on hardware without the need for dedicated INT6 tensor cores, which costs much more than SQ-format. SQ-format has stability advantages on larger models (Llama-3-70B and Qwen-3-30B) and more obvious improvements in generative accuracies. Since the W(SQ)A4 configuration requires dedicated hardware support as shown in Figure 2a, we currently use GPU simulation while do not provide latency comparison.

## 3.3 FINDING 2: SQ-FORMAT SUPPORTS STATIC ACTIVATION QUANTIZATION

The dynamic nature of applying SQ-format to activations on GPUs requires using TopK operations on CUDA cores, which impacts actual throughput without dedicated accelerators. Fortunately, the static strategy can achieve similar performance. We compare the dynamic and static strategies, and the benchmarking results are shown in Table 2.

Comparing the dynamic and static strategy results under different models and configurations, we find static strategy can successfully retain most of the performance. The average benchmark accuracy varies within ±1%. In fact, the dynamic strategy still relys solely on the absolute activations. Although the static strategy is only trained on the calibration set, considering the activation-weight product may compensate for this. In Appendix A.1, we will also show that the static strategy is not

Table 2: Performance comparison of dynamic v.s. static SQ-format on activations. For each setting, **Red bold values** highlight the best performance for static strategy, and **Black bold values** highlight the best performance for dynamic strategy.

| Model | Setting | Static | $b$ | Non-generative Acc (%) | | | | | | Generative Acc (%) | | Perplexity | |
|---|---|---|---|---|---|---|---|---|---|---|---|---|---|
| | | | | ARC-c (↑) | ARC-e (↑) | PIQA (↑) | Hellas. (↑) | Wino. (↑) | OBQA (↑) | GSM8k (↑) | AGIEval (↑) | Wiki. (↓) | Lamb. (↓) |
| Llama-3 70B | BF16 | - | - | 64.51 | 86.11 | 84.39 | 84.96 | 80.19 | 48.40 | 81.05 | 45.67 | 2.91 | 2.58 |
| | W4A(SQ6) $b$-(8/4)-0.5 | ✓ | 16 | **63.73** | **84.51** | **83.89** | **84.42** | 80.03 | **48.20** | **78.77** | **44.07** | **3.43** | **2.64** |
| | | | 32 | 62.37 | 83.67 | 83.46 | 84.06 | 79.63 | 47.40 | 77.93 | 43.86 | 3.57 | 2.67 |
| | | | 64 | 63.56 | 83.54 | 83.13 | 84.35 | **80.26** | 47.60 | 78.24 | **44.07** | 3.74 | **2.64** |
| | | - | 16 | **62.80** | 84.81 | **84.06** | **84.44** | 80.19 | 48.00 | **78.32** | 44.68 | **3.31** | 2.65 |
| | | | 32 | 62.71 | 84.39 | 84.00 | 84.32 | **80.27** | **48.60** | 78.09 | 44.97 | 3.39 | 2.65 |
| | | | 64 | **62.80** | 83.88 | 83.62 | 84.34 | 79.79 | **48.60** | 77.71 | **44.99** | 3.48 | **2.62** |
| | W4A(SQ5) $b$-(8/4)-0.75 | ✓ | 16 | **62.37** | 83.92 | 83.24 | **84.20** | **79.95** | 48.00 | 77.78 | 43.62 | **3.55** | **2.67** |
| | | | 32 | 61.77 | 83.88 | **83.40** | 84.06 | 79.79 | 46.60 | **77.86** | **43.92** | 3.75 | 2.70 |
| | | | 64 | 61.51 | **84.04** | 83.18 | 83.99 | 78.92 | **48.40** | 76.49 | 43.10 | 3.99 | **2.67** |
| | | - | 16 | **62.88** | 84.47 | **83.79** | **84.46** | 79.79 | **48.20** | 79.15 | 44.01 | **3.35** | 2.66 |
| | | | 32 | 61.95 | **84.51** | 83.73 | 84.14 | **80.90** | 47.80 | **79.23** | 44.67 | 3.44 | 2.66 |
| | | | 64 | 62.63 | 83.84 | 83.57 | 84.25 | 80.27 | 48.00 | 77.48 | **44.68** | 3.54 | **2.62** |
| Qwen-3 30B-A3B | BF16 | - | - | 55.80 | 79.17 | 80.58 | 77.65 | 70.88 | 45.80 | 90.45 | 63.05 | 10.88 | 4.11 |
| | W4A(SQ6) $b$-(8/4)-0.5 | ✓ | 16 | **55.46** | **78.62** | **80.14** | **76.66** | **71.59** | **45.80** | 88.10 | **60.06** | **11.15** | **4.34** |
| | | | 32 | 54.18 | 78.45 | 79.43 | 76.46 | 68.82 | 43.40 | **89.01** | 59.83 | 11.34 | 4.37 |
| | | | 64 | 53.33 | 77.44 | 79.71 | 76.46 | 68.59 | 45.20 | 87.49 | 58.38 | 11.31 | 4.60 |
| | | - | 16 | 55.38 | **79.21** | 79.54 | 76.60 | **70.56** | 44.80 | 88.78 | **61.33** | **11.11** | 4.47 |
| | | | 32 | **55.63** | 78.49 | **79.87** | 76.43 | 69.61 | **45.60** | **89.39** | 60.55 | 11.15 | 4.39 |
| | | | 64 | 52.82 | 76.64 | 79.05 | **76.73** | 69.14 | 45.20 | 88.55 | 60.78 | 11.22 | **4.21** |
| | W4A(SQ5) $b$-(8/4)-0.75 | ✓ | 16 | 54.69 | 77.44 | 79.16 | **76.52** | **70.40** | **45.40** | 88.40 | **59.65** | **11.21** | **4.42** |
| | | | 32 | **55.20** | **78.66** | **79.22** | 76.18 | 69.19 | 42.60 | **88.48** | 57.98 | 11.42 | 4.52 |
| | | | 64 | 52.90 | 76.39 | 78.67 | 75.79 | 67.64 | 44.00 | 88.10 | 57.39 | 11.41 | 4.66 |
| | | - | 16 | **55.46** | **78.75** | **80.25** | 76.28 | **70.24** | 44.20 | **89.84** | **61.31** | **11.10** | 4.50 |
| | | | 32 | 55.03 | 78.70 | 79.98 | **76.47** | 69.38 | 44.40 | 88.86 | 60.49 | 11.17 | 4.35 |
| | | | 64 | 52.90 | 76.68 | 78.89 | 76.42 | 68.75 | **45.00** | 87.64 | 60.10 | 11.21 | **4.25** |

particularly dependent on large calibration set size. The grid experiments we conduct on different calibration set sizes show a relatively stable accuracy trend.

# 4 HARDWARE-ALGORITHM CO-DESIGN & DESIGN EXPLORATION

As a format requires dedicated hardware support to achieve most theoretical benefits, SQ-format will benefit from hardware-algorithm co-design. This section discusses how various parameters operate in PTQ process and suggests best practices for hardware implementation based on simulation results.

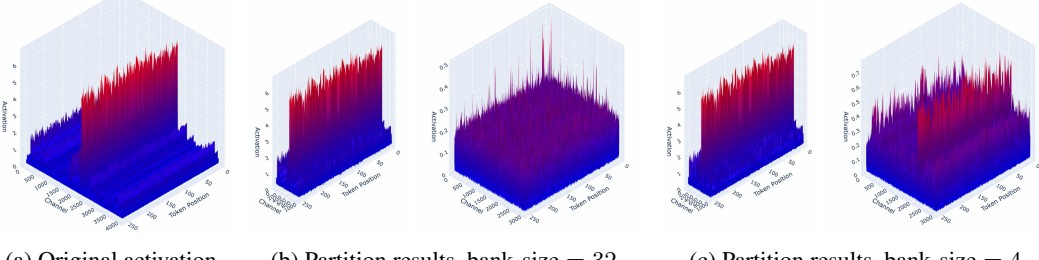

(a) Original activation.  (b) Partition results, bank_size = 32.  (c) Partition results, bank_size = 4.

Figure 6: Activation partition results from Layer 30 of Llama-3-8B. bank_size = 32 provides more flexible precision partition than bank_size = 4.

## 4.1 BANK SIZE & SPARSITY

To facilitate hardware implementation, SQ-format splits the operand matrices into banks, and each bank performs a fixed sparsity precision partitioning. This is to model the numerical importance distribution of the original matrix. However, from the algorithm perspective, this requires careful

setting of bank size and sparsity. We provide a visualization example of static SQ-format on activations in Figure 6. The original matrix has unevenly distributed per-channel activations. Under $s = 3/4$ (4x sparse), $bank\_size = 32$ can successfully separate most important activations, with the low-precision parts appearing to be flat; $bank\_size = 4$ is less flexible, and there are still per-channel outliers in the low-precision part. This resulted in NVIDIA 2:4 semi-structure sparse, implemented with SparseGPT achieving poor results in Table 1.

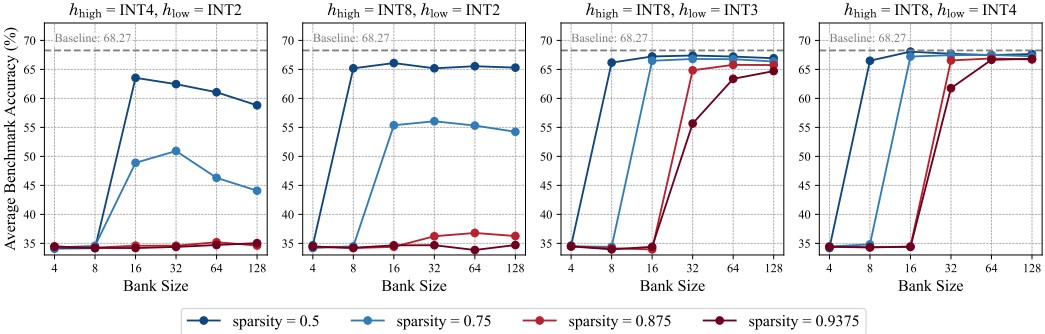

Figure 7: Average benchmark accuracy of using SQ-format on weights under different $h_{high}, h_{low}$, bank_size and sparsity on Llama-3-8B. The BF16 baseline result is dashlined in gray.

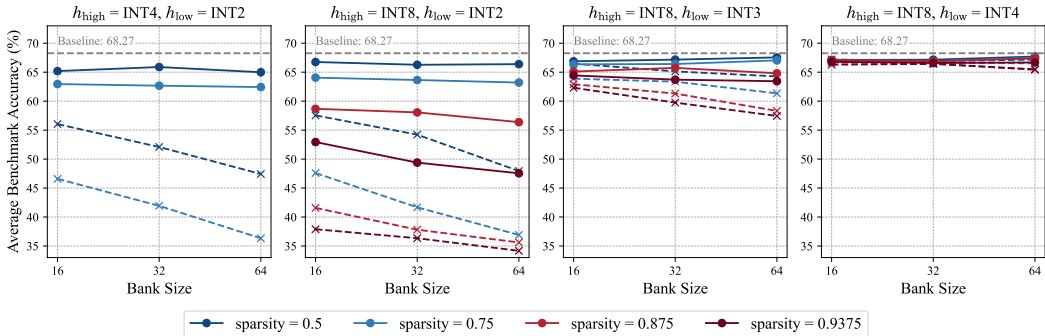

Figure 8: Average benchmark accuracy of using SQ-format on activations under different $h_{high}, h_{low}$, bank_size and sparsity on Llama-3-8B. The BF16 baseline result is dashlined in gray. Dynamic strategy results are marked with ∘ while static results with ×.

We evaluate the performance of different combinations of bank_size and sparsity, as shown in Figure 7. For using SQ-format on weights, under a fixed sparsity, there exists an optimal bank_size as it changes. The larger the sparsity, the larger the optimal bank_size. As illustrated in Figure 8, when using SQ-format on activations, the trend is not as obvious as on weights, showing slow changes. Empirically, static strategy tends to prefer a smaller bank_size. These results guide the design of dedicated hardware. To support at least 16x sparsity for weights, the bank_size illustrated in Figure 2a needs to be at least 64, which affects the area of MUXs. As for activations, the dedicated unit illustraed in Figure 4 should support bank_size 64 or lower.

## 4.2 High/Low Precision Choice & Sparsity

In Figures 7 and 8, we show results of other $h_{high}, h_{low}$ configurations including $(4/2), (8/2), (8/3)$.

**SQ-format support for lower precisions.** For $h_{low} = $ INT2, SQ-format on weights can hardly maintain accuracy, and only with the $B - (8/2) - 0.5$ setting can achieve usable performance. This results from insufficient width of low-precision bits, even introducing high-precision elements is difficult to compensate for the loss. Dynamic strategy for SQ-format on activations is better than that for weights, and it supports up to 4x sparsity. For $h_{low} = $ INT3 and using SQ-format on

weights, $(8/3)$ setting shows similar performance trends to $(8/4)$. There is an opportunity to achieve a tradeoff between storage and performance.

**Computation balance for sparsity.** Sparsity is restricted by $(h_{\text{high}}/h_{\text{low}})$ configuration, which is due to differences in tensor core computational power. For example, using SQ-format on weights, we hope that the computation of the sparse high-precision part can be masked by the low-precision part. Assuming the hardware's computational power for W8A8 is four times that of W4A4, the sparsity needs to be at least 0.75. Therefore, the W(SQ$x$)A$y$ results shown in Table 1 are at least 4x sparsity. When the high-precision remains unchanged, using a lower low-precision such as INT2 would require an even higher minimum sparsity.

## 5 RELATED WORKS

**Post-Training Quantization** converts the weights and/or activations of a model to low-precision data types after the model has completed full-precision pre-training. GPTQ (Frantar et al., 2022a) quantize the weights column by column, successfully compressing OPT-175B to 4 bits. AWQ (Lin et al., 2023) and Wanda (Sun et al., 2024) consider the interaction with activations when quantizing weights. SpQR (Dettmers et al., 2024) and QUIK (Ashkboos et al., 2023) isolate outliers, but they often face hardware efficiency challenges due to their reliance on unstructured or irregular patterns. SmoothQuant (Xiao et al., 2023) is a joint quantization method for weights and activations, which achieves W8A8 PTQ by transferring the difficulty of activating quantization to the weights. SparseGPT (Frantar & Alistarh, 2023) formulates pruning as a series of layer-wise sparse regression problems, which can prune OPT-175B by >50% without significantly sacrificing performance.

**Data Representations for Quantization.** Quantization methods are supported by different data formats. Standard INT8 format (Dettmers et al., 2022) benefits from extensive hardware support and high computing throughput. FP8 format (Micikevicius et al., 2022) provides a wider dynamic range to capture activation outliers. NVFP4 (Alvarez et al., 2025) uses microblock scaling to achieve extreme 4-bit compression with NVIDIA Blackwell hardware. MX formats (Rouhani et al., 2023) adapt to the local data range through shared scaling. HiFloat8 (Luo et al., 2024) features tapered precision and better balances precision and dynamic range. Current formats apply a single precision scheme to all values, which creates a fundamental conflict representing non-uniform information.

## 6 CONCLUSION

We propose SQ-format, a novel data format enabling hybrid-precision computation. By sparsely mixing high-precision elements in a hardware-friendly manner during low-precision computation, SQ-format successfully achieves a Pareto improvement between precision and throughput. We also proposes a static activation quantization algorithm, effectively showing its potential for hardware deployment. We not only provides a practical solution for accelerating LLMs on current hardware but also offers a promising blueprint for the co-design of next-generation AI accelerators.

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

# A COMPLETE EXPERIMENT RESULTS

## A.1 COMPLETE RESULTS FOR PERFORMANCE COMPARISON

Table 3: Performance comparison of dynamic v.s. static SQ-format on activations. The quantized model is Llama-2-7B.

| Model | Setting | Static | Sparsity | $y$ | Non-generative Acc (%) | | | | | | Generative Acc (%) | | Perplexity | |
|---|---|---|---|---|---|---|---|---|---|---|---|---|---|---|
| | | | | | ARC-c (↑) | ARC-e (↑) | PIQA (↑) | Hellas. (↑) | Wino. (↑) | OBQA (↑) | GSM8k (↑) | AGIEval (↑) | Wiki. (↓) | Lamb. (↓) |
| Llama-2 7B | W4A8 | - | 0 | 8 | 44.88 | 74.92 | 78.78 | 75.58 | 68.98 | 44.40 | 14.56 | 28.42 | 8.93 | 3.54 |
| | W4A(SQ$y$) 16-(8/4) | - | 0.5 | 6 | 44.88 | 74.83 | 78.78 | 75.48 | 68.82 | 44.20 | 13.87 | 28.77 | 8.94 | 3.54 |
| | | | 0.75 | 5 | 44.80 | 74.62 | 78.94 | 75.62 | 68.75 | 43.60 | 12.96 | 28.58 | 8.96 | 3.56 |
| | | | 0.875 | 4.5 | 44.88 | 74.58 | 78.73 | 75.39 | 69.06 | 43.80 | 13.19 | 28.80 | 8.99 | 3.58 |
| | | | 0.9375 | 4.25 | 44.45 | 73.86 | 78.89 | 75.11 | 67.96 | 45 | 14.10 | 29.04 | 9.03 | 3.56 |
| | | ✓ | 0.5 | 6 | 44.71 | 74.16 | 78.94 | 75.40 | 69.14 | 44.80 | 12.05 | 28.52 | 8.99 | 3.55 |
| | | | 0.75 | 5 | 44.62 | 74.33 | 78.07 | 75.21 | 68.75 | 44.00 | 12.28 | 28.37 | 9.05 | 3.60 |
| | | | 0.875 | 4.5 | 44.62 | 73.91 | 78.29 | 75.13 | 68.82 | 44.40 | 12.36 | 28.22 | 9.10 | 3.64 |
| | | | 0.9375 | 4.25 | 43.77 | 73.32 | 78.29 | 74.96 | 67.72 | 42.80 | 13.12 | 28.23 | 9.12 | 3.63 |
| | W4A(SQ$y$) 16-(8/3) | - | 0.5 | 5.5 | 44.97 | 74.75 | 79.00 | 75.48 | 69.38 | 44.20 | 14.25 | 28.59 | 8.97 | 3.56 |
| | | | 0.75 | 4.25 | 44.11 | 73.86 | 78.02 | 75.18 | 67.64 | 44.40 | 12.51 | 28.96 | 9.08 | 3.62 |
| | | | 0.875 | 3.625 | 43.86 | 73.36 | 78.18 | 74.85 | 68.51 | 43.20 | 12.21 | 28.36 | 9.28 | 3.75 |
| | | | 0.9375 | 3.3125 | 41.89 | 72.26 | 78.07 | 73.98 | 68.27 | 42.20 | 10.84 | 27.99 | 9.50 | 3.84 |
| | | ✓ | 0.5 | 5.5 | 44.11 | 73.78 | 77.75 | 74.88 | 66.69 | 43.80 | 12.13 | 28.40 | 9.25 | 3.69 |
| | | | 0.75 | 4.25 | 42.83 | 71.93 | 77.58 | 73.91 | 65.98 | 41.80 | 10.77 | 28.37 | 9.59 | 3.91 |
| | | | 0.875 | 3.625 | 43.09 | 70.66 | 77.48 | 73.12 | 66.54 | 40.00 | 10.39 | 28.00 | 9.83 | 4.05 |
| | | | 0.9375 | 3.3125 | 42.75 | 71.51 | 77.69 | 73.22 | 65.35 | 40.60 | 9.02 | 27.18 | 9.99 | 4.08 |
| | W4A(SQ$y$) 16-(8/2) | - | 0.5 | 5 | 43.69 | 74.16 | 78.94 | 74.81 | 68.19 | 43.80 | 13.04 | 28.91 | 9.12 | 3.61 |
| | | | 0.75 | 3.5 | 41.81 | 70.88 | 77.37 | 73.26 | 64.96 | 42.00 | 10.08 | 27.93 | 9.85 | 3.96 |
| | | | 0.875 | 2.75 | 38.48 | 67.09 | 76.01 | 70.58 | 64.09 | 38.20 | 5.84 | 27.43 | 11.30 | 4.88 |
| | | | 0.9375 | 2.375 | 35.92 | 62.12 | 73.67 | 66.14 | 59.12 | 36.40 | 2.73 | 26.23 | 13.75 | 6.56 |
| | | ✓ | 0.5 | 5 | 35.75 | 63.97 | 73.94 | 66.68 | 63.69 | 37.80 | 6.22 | 27.04 | 12.36 | 4.57 |
| | | | 0.75 | 3.5 | 31.06 | 54.38 | 69.21 | 56.07 | 57.77 | 34.00 | 2.20 | 25.79 | 19.07 | 8.68 |
| | | | 0.875 | 2.75 | 29.01 | 48.91 | 65.61 | 49.71 | 55.72 | 31.60 | 1.74 | 25.16 | 27.45 | 15.91 |
| | | | 0.9375 | 2.375 | 27.73 | 45.41 | 63.76 | 46.13 | 53.20 | 31.80 | 1.97 | 25.47 | 36.87 | 31.20 |
| | W4A(SQ$y$) 16-(4/2) | - | 0.5 | 3 | 43.26 | 73.78 | 78.45 | 74.29 | 67.88 | 42.60 | 11.98 | 28.45 | 9.26 | 3.65 |
| | | | 0.75 | 2.5 | 43.43 | 71.09 | 76.61 | 72.66 | 65.51 | 40.00 | 10.84 | 27.26 | 9.93 | 4.00 |
| | | ✓ | 0.5 | 3 | 36.86 | 62.25 | 74.65 | 66.31 | 63.93 | 37.40 | 4.62 | 26.62 | 12.58 | 4.74 |
| | | | 0.75 | 2.5 | 30.46 | 53.66 | 68.55 | 56 | 58.41 | 34.20 | 2.27 | 25.76 | 19.21 | 9.11 |

Table 4: Performance comparison of quantization settings. The quantized model is Llama-3-8B.

| Model | Setting | Static | $b$ | Non-generative Acc (%) | | | | | | Generative Acc (%) | | Perplexity | |
|---|---|---|---|---|---|---|---|---|---|---|---|---|---|
| | | | | ARC-c (↑) | ARC-e (↑) | PIQA (↑) | Hellas. (↑) | Wino. (↑) | OBQA (↑) | GSM8k (↑) | AGIEval (↑) | Wiki. (↓) | Lamb. (↓) |
| Llama-3 8B | W16A16 | - | - | 53.41 | 77.74 | 80.85 | 79.21 | 73.40 | 45.00 | 49.43 | 33.68 | 7.26 | 3.09 |
| | W(SQ8)A4 $b$-(8/4)-0.5 | - | 16 | 51.79 | 77.19 | 80.03 | 78.23 | 72.85 | 44.60 | 42.68 | 32.76 | 7.75 | 3.34 |
| | | | 32 | 50.51 | 77.27 | 78.67 | 77.77 | 70.96 | 44.60 | 41.93 | 31.49 | 7.94 | 3.40 |
| | | | 64 | 48.98 | 73.70 | 77.53 | 77.23 | 71.19 | 42.40 | 39.27 | 31.58 | 8.23 | 3.60 |
| | W(SQ6)A4 $b$-(8/4)-0.75 | - | 16 | 47.70 | 76.26 | 79.16 | 76.74 | 72.38 | 43.80 | 36.16 | 31.66 | 8.34 | 3.62 |
| | | | 32 | 49.40 | 75.42 | 79.38 | 77.75 | 71.74 | 44.00 | 39.58 | 31.91 | 8.01 | 3.42 |
| | | | 64 | 47.53 | 73.48 | 78.07 | 77.06 | 71.27 | 44.80 | 37.83 | 31.02 | 8.32 | 3.59 |
| | W(SQ8)A8 $b$-(8/4)-0.5 | - | 16 | 52.65 | 77.53 | 80.85 | 79.10 | 72.93 | 45.80 | 48.60 | 33.84 | 7.31 | 3.13 |
| | | | 32 | 52.56 | 77.48 | 80.85 | 79.29 | 73.01 | 44.40 | 48.14 | 33.43 | 7.28 | 3.10 |
| | | | 64 | 53.41 | 78.07 | 80.79 | 79.19 | 73.09 | 44.80 | 48.29 | 33.93 | 7.29 | 3.09 |
| | W(SQ6)A8 $b$-(8/4)-0.75 | - | 16 | 50.94 | 76.77 | 80.69 | 77.99 | 73.01 | 45.40 | 40.11 | 32.24 | 7.87 | 3.38 |
| | | | 32 | 53.84 | 77.15 | 80.96 | 79.05 | 72.77 | 44.40 | 47.01 | 33.54 | 7.37 | 3.12 |
| | | | 64 | 52.90 | 77.74 | 80.03 | 79.23 | 71.90 | 44.40 | 45.87 | 33.54 | 7.35 | 3.12 |
| | W4A(SQ6) $b$-(8/4)-0.5 | ✓ | 16 | 49.83 | 75.25 | 80.14 | 78.03 | 73.16 | 46.80 | 40.49 | 32.93 | 7.83 | 3.36 |
| | | | 32 | 50.34 | 75.72 | 79.76 | 77.93 | 72.85 | 44.80 | 40.41 | 32.12 | 7.99 | 3.34 |
| | | | 64 | 51.96 | 76.94 | 79.60 | 77.19 | 73.32 | 44.00 | 37.91 | 31.82 | 8.18 | 3.53 |
| | | - | 16 | 50.00 | 75.67 | 80.09 | 78.33 | 73.48 | 45.00 | 39.35 | 32.87 | 7.74 | 3.36 |
| | | | 32 | 51.02 | 76.05 | 80.63 | 77.94 | 72.93 | 44.40 | 42.38 | 32.73 | 7.84 | 3.30 |
| | | | 64 | 52.13 | 77.44 | 79.87 | 78.05 | 73.48 | 45.20 | 40.26 | 32.35 | 7.97 | 3.47 |
| | W4A(SQ5) $b$-(8/4)-0.75 | ✓ | 16 | 50.00 | 74.96 | 79.92 | 77.76 | 73.32 | 45.80 | 38.36 | 32.58 | 7.93 | 3.40 |
| | | | 32 | 49.74 | 75.04 | 80.30 | 77.58 | 72.53 | 44.40 | 41.85 | 31.92 | 8.13 | 3.43 |
| | | | 64 | 50.26 | 76.73 | 79.33 | 76.58 | 71.51 | 44.80 | 38.67 | 31.06 | 8.39 | 3.66 |
| | | - | 16 | 49.74 | 75.84 | 79.43 | 78.04 | 73.16 | 45.20 | 39.95 | 32.75 | 7.79 | 3.39 |
| | | | 32 | 50.43 | 76.09 | 80.20 | 77.93 | 72.61 | 44.00 | 41.55 | 32.46 | 7.88 | 3.31 |
| | | | 64 | 51.96 | 77.15 | 79.43 | 77.68 | 74.27 | 44.40 | 40.94 | 32.35 | 8.02 | 3.50 |

Table 5: Performance comparison of quantization settings. The quantized model is Llama-3-70B.

| Model | Setting | Static | $b$ | Non-generative Acc (%) | | | | | | Generative Acc (%) | | Perplexity | |
| | | | | ARC-c (↑) | ARC-e (↑) | PIQA (↑) | Hellas. (↑) | Wino. (↑) | OBQA (↑) | GSM8k (↑) | AGIEval (↑) | Wiki. (↓) | Lamb. (↓) |
|---|---|---|---|---|---|---|---|---|---|---|---|---|---|
| | W16A16 | - | - | 64.51 | 86.11 | 84.39 | 84.96 | 80.19 | 48.40 | 81.05 | 45.67 | 2.92 | 2.58 |
| | W(SQ8)A4 $b$-(8/4)-0.5 | - | 16 | 61.26 | 83.50 | 83.68 | 84.20 | 79.87 | 47.40 | 79.61 | 42.56 | 3.68 | 2.69 |
| | | | 32 | 61.35 | 83.38 | 83.41 | 83.68 | 78.77 | 47.60 | 77.56 | 42.83 | 3.89 | 2.70 |
| | | | 64 | 59.64 | 81.73 | 82.59 | 83.51 | 77.19 | 47.00 | 75.66 | 40.71 | 4.23 | 2.82 |
| | W(SQ6)A4 $b$-(8/4)-0.75 | - | 16 | 26.37 | 25.00 | 51.31 | 26.84 | 56.20 | 29.80 | 0.00 | 24.42 | > 100 | > 100 |
| | | | 32 | 61.01 | 83.12 | 83.03 | 84.10 | 78.77 | 46.40 | 78.62 | 41.84 | 3.79 | 2.66 |
| | | | 64 | 60.41 | 81.86 | 83.24 | 83.98 | 77.35 | 48.20 | 74.91 | 41.38 | 4.07 | 2.70 |
| | W(SQ8)A8 $b$-(8/4)-0.5 | - | 16 | 64.33 | 85.94 | 84.43 | 84.79 | 80.18 | 48.60 | 79.98 | 45.28 | 2.95 | 2.58 |
| | | | 32 | 64.59 | 85.90 | 84.38 | 84.86 | 80.82 | 49.00 | 80.13 | 45.46 | 2.93 | 2.59 |
| | | | 64 | 64.41 | 85.81 | 84.49 | 84.83 | 80.82 | 48.00 | 80.59 | 45.47 | 2.93 | 2.58 |
| Llama-3 70B | W(SQ6)A8 $b$-(8/4)-0.75 | - | 16 | 63.39 | 85.05 | 83.89 | 83.94 | 80.03 | 48.00 | 79.37 | 43.05 | 3.42 | 2.68 |
| | | | 32 | 64.24 | 85.81 | 84.27 | 84.85 | 80.42 | 48.80 | 80.81 | 45.94 | 2.99 | 2.58 |
| | | | 64 | 63.65 | 86.23 | 84.11 | 84.96 | 80.18 | 48.40 | 79.98 | 46.08 | 2.98 | 2.59 |
| | W4A(SQ6) $b$-(8/4)-0.5 | ✓ | 16 | 63.73 | 84.51 | 83.89 | 84.42 | 80.03 | 48.20 | 78.77 | 43.51 | 3.43 | 2.67 |
| | | | 32 | 62.37 | 83.67 | 83.46 | 84.06 | 79.63 | 47.40 | 77.93 | 43.86 | 3.58 | 2.67 |
| | | | 64 | 63.56 | 83.54 | 83.13 | 84.35 | 80.26 | 47.60 | 78.24 | 44.07 | 3.75 | 2.64 |
| | | - | 16 | 62.80 | 84.81 | 84.06 | 84.44 | 80.19 | 48.00 | 78.32 | 44.68 | 3.31 | 2.65 |
| | | | 32 | 62.71 | 84.39 | 84.00 | 84.32 | 80.27 | 48.60 | 78.09 | 44.97 | 3.40 | 2.65 |
| | | | 64 | 62.80 | 83.88 | 83.62 | 84.34 | 79.79 | 48.60 | 77.71 | 44.99 | 3.49 | 2.62 |
| | W4A(SQ5) $b$-(8/4)-0.75 | ✓ | 16 | 62.37 | 83.92 | 83.24 | 84.20 | 79.95 | 48.00 | 77.78 | 43.62 | 3.56 | 2.68 |
| | | | 32 | 61.77 | 83.88 | 83.40 | 84.06 | 79.79 | 46.60 | 77.86 | 43.92 | 3.76 | 2.70 |
| | | | 64 | 61.51 | 84.04 | 83.18 | 83.99 | 78.92 | 48.40 | 76.49 | 43.10 | 4.00 | 2.68 |
| | | - | 16 | 62.88 | 84.47 | 83.79 | 84.46 | 79.79 | 48.20 | 79.15 | 44.01 | 3.36 | 2.66 |
| | | | 32 | 61.95 | 84.51 | 83.73 | 84.14 | 80.90 | 47.80 | 79.23 | 44.67 | 3.45 | 2.67 |
| | | | 64 | 62.63 | 83.84 | 83.57 | 84.25 | 80.27 | 48.00 | 77.48 | 44.68 | 3.55 | 2.63 |

Table 6: Performance comparison of quantization settings. The quantized model is Qwen3-30B-A3B.

| Model | Setting | Static | $b$ | Non-generative Acc (%) | | | | | | Generative Acc (%) | | Perplexity | |
| | | | | ARC-c (↑) | ARC-e (↑) | PIQA (↑) | Hellas. (↑) | Wino. (↑) | OBQA (↑) | GSM8k (↑) | AGIEval (↑) | Wiki. (↓) | Lamb. (↓) |
|---|---|---|---|---|---|---|---|---|---|---|---|---|---|
| | W16A16 | - | - | 55.80 | 79.17 | 80.58 | 77.65 | 70.88 | 45.80 | 90.45 | 63.05 | 10.89 | 4.12 |
| | W(SQ8)A4 $b$-(8/4)-0.5 | - | 16 | 54.44 | 76.94 | 79.38 | 76.20 | 69.06 | 43.60 | 89.23 | 58.84 | 11.14 | 4.35 |
| | | | 32 | 52.30 | 75.97 | 79.00 | 75.92 | 69.30 | 42.20 | 87.79 | 57.38 | 11.35 | 4.59 |
| | | | 64 | 53.75 | 75.97 | 79.38 | 75.72 | 67.40 | 42.00 | 87.34 | 54.95 | 11.57 | 5.11 |
| | W(SQ6)A4 $b$-(8/4)-0.75 | - | 16 | 55.38 | 77.78 | 79.22 | 75.03 | 68.51 | 43.00 | 5.23 | 52.23 | 17.90 | 5.98 |
| | | | 32 | 53.41 | 77.15 | 79.38 | 75.82 | 68.43 | 43.40 | 87.79 | 56.87 | 11.25 | 4.45 |
| | | | 64 | 53.75 | 76.22 | 78.02 | 75.58 | 70.64 | 44.40 | 88.17 | 56.57 | 11.43 | 4.86 |
| | W(SQ8)A8 $b$-(8/4)-0.5 | - | 16 | 55.55 | 77.95 | 80.09 | 76.70 | 69.85 | 42.60 | 89.76 | 61.74 | 11.01 | 4.29 |
| | | | 32 | 55.80 | 79.04 | 80.79 | 76.87 | 69.61 | 44.00 | 89.39 | 61.37 | 11.09 | 4.30 |
| | | | 64 | 54.27 | 77.74 | 80.03 | 77.04 | 69.69 | 44.20 | 88.17 | 60.63 | 10.98 | 4.25 |
| Qwen-3 30B-A3B | W(SQ6)A8 $b$-(8/4)-0.75 | - | 16 | 55.03 | 78.41 | 79.16 | 75.80 | 69.69 | 43.40 | 58.00 | 57.51 | 14.17 | 4.57 |
| | | | 32 | 55.55 | 78.70 | 79.71 | 77.12 | 71.11 | 43.20 | 89.01 | 61.62 | 11.08 | 4.23 |
| | | | 64 | 54.95 | 78.45 | 79.87 | 77.00 | 70.17 | 43.40 | 88.32 | 62.32 | 11.00 | 4.11 |
| | W4A(SQ6) $b$-(8/4)-0.5 | ✓ | 16 | 55.46 | 78.62 | 80.14 | 76.66 | 71.59 | 45.80 | 88.10 | 60.06 | 11.16 | 4.35 |
| | | | 32 | 54.18 | 78.45 | 79.43 | 76.46 | 68.82 | 43.40 | 89.01 | 59.83 | 11.35 | 4.38 |
| | | | 64 | 53.33 | 77.44 | 79.71 | 76.46 | 68.59 | 45.20 | 87.49 | 58.38 | 11.31 | 4.61 |
| | | - | 16 | 55.38 | 79.21 | 79.54 | 76.60 | 70.56 | 44.80 | 88.78 | 61.33 | 11.11 | 4.47 |
| | | | 32 | 55.63 | 78.49 | 79.87 | 76.43 | 69.61 | 45.60 | 89.39 | 60.55 | 11.16 | 4.40 |
| | | | 64 | 52.82 | 76.64 | 79.05 | 76.73 | 69.14 | 45.20 | 88.55 | 60.78 | 11.22 | 4.22 |
| | W4A(SQ5) $b$-(8/4)-0.75 | ✓ | 16 | 54.69 | 77.44 | 79.16 | 76.52 | 70.40 | 45.40 | 88.40 | 59.65 | 11.22 | 4.43 |
| | | | 32 | 55.20 | 78.66 | 79.22 | 76.18 | 69.14 | 42.60 | 88.48 | 57.98 | 11.42 | 4.53 |
| | | | 64 | 52.90 | 76.39 | 78.67 | 75.79 | 67.64 | 44.00 | 88.10 | 57.39 | 11.41 | 4.67 |
| | | - | 16 | 55.46 | 78.75 | 80.25 | 76.28 | 70.24 | 44.20 | 89.84 | 61.31 | 11.10 | 4.51 |
| | | | 32 | 55.03 | 78.70 | 79.98 | 76.47 | 69.38 | 44.40 | 88.86 | 60.49 | 11.18 | 4.35 |
| | | | 64 | 52.90 | 76.68 | 78.89 | 76.42 | 68.75 | 45.00 | 87.64 | 60.10 | 11.22 | 4.26 |

## A.2 IMPACT CALIBRATE SET SIZE

For static strategy for SQ-format on activations, it is worth noting whether a small amount of calibration sets can make the precision masks have enough generalization. To this end, we conduct

experiments with different calibration set sizes, as shown in Table 7. The conclusion shows that the performance is relatively stable when using different calibration set sizes, indicating that the outlier distribution of the activations can be easily modeled.

Table 7: Performance comparison of calibration sizes.

| Model | Setting | Calib. Size | Non-generative Acc (%) | | | | | | Generative Acc (%) | | Perplexity | |
|---|---|---|---|---|---|---|---|---|---|---|---|---|
| | | | ARC-c (↑) | ARC-e (↑) | PIQA (↑) | Hellas. (↑) | Wino. (↑) | OBQA (↑) | GSM8k (↑) | AGIEval (↑) | Wiki. (↓) | Lamb. (↓) |
| | W16A16 | - | 53.41 | 77.74 | 80.85 | 79.21 | 73.40 | 45.00 | 49.43 | 33.68 | 7.25 | 3.09 |
| Llama-3 8B | W4A(SQ6) 16-(8/4)-0.5 | 8 | 52.39 | 78.24 | 80.03 | 77.60 | 73.40 | 45.60 | 39.50 | 32.80 | 7.84 | 3.32 |
| | | 16 | 50.09 | 76.22 | 80.20 | 78.10 | 73.24 | 45.20 | 39.50 | 31.98 | 7.83 | 3.38 |
| | | 32 | 49.83 | 75.25 | 80.14 | 78.03 | 73.16 | 46.80 | 40.49 | 32.93 | 7.82 | 3.35 |
| | | 64 | 50.34 | 76.26 | 79.87 | 77.98 | 72.14 | 45.00 | 40.71 | 32.74 | 7.84 | 3.35 |
| | | 128 | 49.83 | 76.26 | 79.33 | 77.76 | 73.24 | 45.60 | 39.35 | 32.35 | 7.82 | 3.27 |
| | | 256 | 52.22 | 76.81 | 79.60 | 77.43 | 73.01 | 46.00 | 41.32 | 33.03 | 7.82 | 3.28 |
| | | 512 | 51.88 | 77.69 | 80.09 | 77.94 | 73.64 | 45.00 | 40.94 | 32.54 | 7.83 | 3.21 |

# B  FP QUANTIZATION RESULTS

To demonstrate the effectiveness of SQ-format on floating point, we also conduct experiments on DeepSeek-R1 (685B) with FP8/FP4 quantization. We apply SQ-format on weights (bank_size = 64, sparsity = 0.875) and keep activations in FP8, achieving an effective bit-width of 5 bits. As shown in Table 8, SQ-format demonstrates strong scalability on massive models, maintaining near-lossless performance compared to the BF16 baseline across both generative and non-generative benchmarks.

Table 8: Performance of SQ-format on DeepSeek-R1.

| Model | Setting | Sparsity | Non-generative Acc (%) | | | | | | Generative Acc (%) | | Perplexity | |
|---|---|---|---|---|---|---|---|---|---|---|---|---|
| | | | ARC-c (↑) | ARC-e (↑) | PIQA (↑) | Hellas. (↑) | Wino. (↑) | OBQA (↑) | GSM8k (↑) | AGIEval (↑) | Wiki. (↓) | Lamb. (↓) |
| DeepSeek-R1 | W16A16 | 0 | 64.42 | 85.52 | 84.98 | 87.43 | 79.95 | 48.47 | 95.83 | 70.22 | 3.33 | 12.67 |
| | W(SQ5)A8 | 0.875 | 63.99 | 85.52 | 85.31 | 87.19 | 79.45 | 46.67 | 96.21 | 69.58 | 3.39 | 12.77 |

# C  HARDWARE SYNTHESIS ANALYSIS

To assess the hardware overhead of the dynamic mask handling in SQ-format, we conduct RTL synthesis experiments. We implement the SQ-format unit designed for 4x sparse weights and INT8 activations. For a fair comparison, we select a standard INT6 MAC array as the baseline.

Table 9 presents the normalized area breakdown. This result validates that SQ-format is a physically efficient design capable of increasing compute density for next-generation AI accelerators.

Table 9: Area synthesis comparison normalized by adder area.

| Component | SQ-format (Ours) | Standard INT6 MAC |
|---|---|---|
| Multiplier | 0.42 | 2.23 |
| Adder | 0.24 | 1.00 |
| Accumulator | 0.21 | 0.002 |
| Low-bit Multiplier | 0.42 | – |
| Gather Unit | 0.783 | – |
| Total Area | 2.073 | 3.232 |

# D  END-TO-END LATENCY ANALYSIS

We profile the end-to-end prefilling latency and effective memory bandwidth of SQ-format with static activation strategy on GPUs. The profiling is conducted using the WikiText dataset.

As shown in Table 10, SQ-format significantly outperforms the W4A8 baseline, effectively bridging the gap towards the theoretical W4A4 performance limit while maintaining accuracy.

Table 10: End-to-end prefilling latency and effective bandwidth on GPUs. Setup: bank_size = 64.

| Model | Method | Sparsity | Time | Effective BW (GB/s) | Speedup |
|---|---|---|---|---|---|
| Llama-3-8B | W4A8 | - | 48s | 34.69 | 1.00× |
| | SQ-format | 0.5 | 45s | 35.75 | 1.07× |
| | SQ-format | 0.75 | 42s | 36.90 | 1.14× |
| | SQ-format | 0.875 | 41s | 38.19 | **1.17×** |
| | W4A4 | - | 38s | 40.61 | 1.26× |
| Llama-3-70B | W4A8 | - | 10m 8s | 10.84 | 1.00× |
| | SQ-format | 0.5 | 7m 42s | 14.27 | 1.32× |
| | SQ-format | 0.75 | 6m 29s | 16.94 | 1.56× |
| | SQ-format | 0.875 | 5m 55s | 18.54 | **1.71×** |
| | W4A4 | - | 5m 16s | 20.86 | 1.92× |

## E  REPRODUCTION STATEMENT

To facilitate future research and ensure the reproducibility of our results, we commit to releasing our PTQ framework, including the source code and the kernels for the static SQ-format on GPUs upon the acceptance of this paper.

**Discussion on Baseline Re-evaluation.** In our experiments, we re-evaluate all baseline methods within a unified environment rather than directly citing numbers from their original papers. This decision is driven by two primary factors: (1) Model and Benchmark Coverage: Many prior PTQ studies conduct evaluations on older models (e.g., OPT, Llama). As shown in our main results, generative tasks (e.g., GSM8k, AGIEval) are often more sensitive to quantization than standard perplexity or non-generative tasks. To validate the robustness of SQ-format, we extend the evaluation scope to these challenging generative settings on modern LLMs (Llama-3, Qwen3). (2) Unified Evaluation Environment: To ensure a fair comparison, it is critical to align the calibration sets, sequence lengths, and evaluation libraries.

We acknowledge that our reproduced results may differ slightly from the values reported in the original papers. A notable example raised during the review process is the performance of SpinQuant on OBQA, as illustrated in Table 11. This discrepancy primarily stem from the metric selection. We consistently prioritize acc_norm over raw acc where available.

Table 11: Accuracy (%) comparison of SpinQuant (W4A8) on Llama-3-8B.

| Source | ARC-e | ARC-c | PIQA | HellaSwag | WinoGrande | OBQA |
|---|---|---|---|---|---|---|
| SpinQuant (Our reproduction) | 75.08 | 51.78 | 80.20 | 77.71 | 74.19 | 44.40 |
| SpinQuant (Original Paper) | 76.50 | 54.00 | 79.60 | 78.10 | 72.40 | 56.40 |

## F  DISCLOSURE OF LLM USAGE

We make limited use of LLMs to aid in polishing writing. All ideas, analyses, data interpretations, and conclusions presented in this paper are our own.

