# OpenReview forum: "SQ-format: A Unified Sparse-Quantized Hardware-friendly Data Format for Large Language Models"
_ICLR.cc/2026/Conference — Submitted to ICLR 2026_

### Official Review · Reviewer_CDnV · 2025-10-29

**Soundness:** 3
**Presentation:** 4
**Contribution:** 2
**Rating:** 2
**Confidence:** 3

**Summary:**

The paper introduces SQ-Format for post-training quantization (PTQ). The format divides a tensor into blocks, and in each block, a portion of elements (controlled by sparsity ratio s) are quantized to a higher precision such as INT8, while the rest are quantized to a lower precision such as INT4. The goal is to leverage the sparsity of high-precision elements and achieve speedups close to full low-precision matmuls. The authors propose two algorithms for compressing weights and activations to SQ-Format. Their accuracy results are promising (both for weights and static/dynamic activations).

**Strengths:**

- The paper is well-written.
- Extensive experiments + good set of baselines.
- Interesting algorithmic contributions, including the static activation masks.

**Weaknesses:**

- The idea is not novel. Quantization + higher precision outlier formats have been considered by multiple works including SpQR (which the paper cites) and QUIK [1].
- The runtime improvement are more like a promise (if such hardware is designed), hence calling the method hardware-friendly in the title is a bit questionable.
- I am not convinced by the speedups. While there seems to be a kernel implemented for the static version, its throughput is only reported on a single matmul.

[1] https://arxiv.org/pdf/2310.09259

**Questions:**

1. Regarding the first two contribution listed in the intro (SQ-Format definition and implementation / Pareto improvement): I believe the definition is not novel, as I mentioned above. Additionally, can the authors provide a clear plot of accuracy vs runtime to show pareto superiority against other baselines (static activations would suffice, as you already have kernels for them)? In the current state, I'm not convinced merely from Tables 1 and 2 that there's pareto improvement.

2. More broadly, can you include speedup numbers in Tables 1 and 2 (for weights and static activations)?

3. Would it be possible to apply the SQ-Format to both operands? What are the challenges? A brief discussion would suffice.

4. The algorithm picks the most "important" elements to keep in higher precision. Aside from importance, looking at the low-precision quantization error could also interesting here. For example, if an important element can be perfectly captured by the low-precision component, then there is no reason to keep it in high precision. As a suggestion, wouldn't a combination of importance and low-precision error potentially achieve better results?

5. Any idea why the static activation strategy prefers smaller bank sizes? To me it seems counter-intuitive.

In general, I believe the paper's novelty is somewhat questionable, and the pareto superiority is not convincing. I would be open to increasing my score, depending the authors' rebuttal regarding novelty and questions 1 and 2.

---

> ### Author Response · Authors · 2025-11-22
>
> Thank you for your feedback and for acknowledging the strengths of our paper! We have summarized the weaknesses and questions and respond as follows:
>
> > **Weakness 1 & Question 1: Not novel ideas & comparison with SpQR and QUIK.**
> >
>
> We thank you for the valuable reference to QUIK, which we will cite and discuss in the revised paper. While SpQR, QUIK, and SQ-format all share the high-level concept of outlier isolation, SQ-format differs fundamentally in its data layout and hardware execution paradigm, addressing specific efficiency bottlenecks found in both prior works:
>
> - **Structured Regularity (vs. SpQR)**: SpQR relies on unstructured data representation, necessitating explicit 16-bit column indices for every outlier and complex runtime load-balancing.
> - **Hardware Generalization (vs. QUIK)**: While QUIK effectively reduces memory traffic using mixed FP16/INT4, it often faces constraints of retaining down-projection layers in 8-bit and lacks a specific architectural constraint for compute acceleration.
>
> In contrast, SQ-format utilizes a bank-based sparsity structure. This structural constraint enforces aligned memory access and ensures deterministic workload distribution. Our hardware design insight and the exploration of optimal bank sizes bridge a significant gap in sparsity features of future AI accelerators.
>
> > **Weakness 2: Skeptical on hardware-friendliness claim.**
> >
>
> We understand your skepticism regarding the hardware-friendliness claim given the reliance on simulation. To move beyond theoretical promises and validate the physical efficiency of our design, we perform RTL synthesis using the TSMC 12nm process library. We compare SQ-format against a standard integer MAC array under an ISO-I/O-bandwidth constraint (12-bit I/O). The baseline is standard INT6 MAC Array (6-bit weight + 6-bit activation). We implement SQ-format MAC with 4x Sparse weight + INT8 activation, effectively matching 12-bit I/O bandwidth.
>
> **Rebuttal Table 1: Area Synthesis Comparison (Normalized by Adder Area).**
>
> | **Area \ Format** | **SQ-format (Ours)** | **Standard INT6 MAC** |
> | --- | --- | --- |
> | Multiplier | 0.42 | 2.23 |
> | Adder | 0.24 | 1 |
> | Accumulator | 0.21 | 0.002 |
> | Low bit mult. | 0.42 | -- |
> | Gather Unit | 0.783 | -- |
> | Total Area | 2.073 | 3.232 |
>
> As shown in Rebuttal Table 1, SQ-format architecture achieves a 35.8% reduction in total silicon area. In hardware design, area reduction is a direct proxy for static and dynamic power savings. This proves that SQ-format is not just a theoretical promise but a physically efficient design that increases compute density for next-generation accelerators.

---

> ### Author Response · Authors · 2025-11-22
>
> > **Weakness 3: No real-world latency & bandwidth data.**
> >
>
> We thank you for raising concerns regarding end-to-end performance. We address these aspects below with real-world profiling data.
>
> We profile our static activation CUDA implementation on GPUs using the WikiText dataset to measure the real-world impact of SQ-format on memory bandwidth utilization and end-to-end latency. The results are in Rebuttal Table 2.
>
> **Rebuttal Table 2: End-to-end prefilling latency & effective bandwidth on GPUs.**
>
> | **Model** | **Type** | **Sparsity** | **Time** | **Effective BW (GB/s)** | **Speedup vs W4A8** |
> | --- | --- | --- | --- | --- | --- |
> | Llama-3-8B | W4A8 | - | 48s | 34.69 | 1x |
> |  | SQ-format | 0.75 | 42s | 36.90 | 1.14x |
> |  | SQ-format | 0.875 | 41s | 38.19 | 1.17x |
> |  | W4A4 | - | 38s | 40.61 | 1.26x |
> | Llama-3-70B | W4A8 | - | 10min8s | 10.84 | 1x |
> |  | SQ-format | 0.75 | 6min29s | 16.94 | 1.56x |
> |  | SQ-format | 0.875 | 5min55s | 18.54 | 1.71x |
> |  | W4A4 | - | 5min16s | 20.86 | 1.92x |
>
> The results show SQ-format significantly reduces inference latency compared to the W4A8 baseline. For Llama-3-70B, SQ-format $(s=0.875)$ achieves a 1.71x speedup, approaching W4A4. For Llama-3-8B, we observe a consistent 1.17x speedup.
>
> We will include these key data in the revised paper to support the theoretical benefits of the SQ-format.
>
> > **Question 1 & 2: Demonstrating Pareto improvement with runtime & speedup**
> >
>
> We thank you for this request.
>
> **Applying SQ-format to weights** relies on the dedicated hardware primitives proposed in Section 2.3. Since these units do not exist in current GPUs, we cannot measure physical speedup directly. However, our hardware synthesis results in Rebuttal Table 1 show a 35.8% area reduction compared to standard baselines, confirming that the theoretical throughput gains are physically realizable on future accelerators.
>
> **For the static activation strategy**, we have measured real-world speedups on GPUs with our kernel. To demonstrate Pareto improvement, we correlate these speedups with the average accuracy results reported in the main paper.
>
> **Rebuttal Table 3: Pareto comparison of speedup vs. average accuracy.**
>
> | **Model** | **Method** | **Setting** | **Speedup (vs. W4A8)** | **Avg. Acc (Non-gen)** | **Avg. Acc (Gen)** |
> | --- | --- | --- | --- | --- | --- |
> | Llama-3-8B | W4A8 | GPTQ | 1x | 67.86% | 36.87% |
> |  | W4A4 | GPTQ | 1.26x | 64.23% | 29.52% |
> |  | SQ-format | $s=0.5$ | 1.07x | 67.69% | 36.31% |
> |  | SQ-format | $s=0.75$ | 1.14x | 67.48% | 36.64% |
> | Llama-3-70B | W4A8 | GPTQ | 1x | 73.86% | 61.66% |
> |  | W4A4 | GPTQ | 1.92x | 71.93% | 57.71% |
> |  | SQ-format | $s=0.5$ | 1.31x | 73.84% | 61.35% |
> |  | SQ-format | $s=0.75$ | 1.56x | 73.76% | 61.08% |
>
> Note: Speedup is derived from end-to-end latency measurements on the WikiText dataset. Avg. Acc is calculated from Tables 1, 2 and 4 in the paper.
>
> The results confirm that SQ-format successfully breaks the binary trade-off between W4A8 (slow, accurate) and W4A4 (fast, inaccurate), offering a Pareto improvement. We will include these key data in the revised paper to support the theoretical benefits of the SQ-format.

---

> ### Author Response · Authors · 2025-11-22
>
> > **Question 3: Would it be possible to apply the SQ-format to both operands?**
> >
>
> We thank you for your insightful question. While SQ-format is highly effective when applied to a single operand, extending it to both operands is **technically possible** but introduces significant challenges. This is equivalent to supporting sparse-sparse matmul, which **will undermine the structured advantages of SQ-format** and introduce critical architectural difficulties:
>
> - At any element position, we cannot guarantee the multiplication will be low $\times$ low or high $\times$ high. Hardware would need complex, real-time control logic to check the precision flag of both operands, leading to significantly increased instruction branching and pipeline complexity.
> - NVIDIA's 2:4 semi-structured sparsity imposes strict structural rules on only one of the matrix multiplication operands precisely to guarantee computational regularity and predictable hardware execution.
>
> > **Question 4: Incorporating low-precision error into outlier selection.**
> >
>
> We thank you for this theoretically interesting suggestion. While combining importance with quantization error is intuitive, we prioritize the current importance-based metric for two reasons rooted in the properties of LLMs and our hardware-friendly design goal.
>
> **For activations**, our goal is to generate a static mask that generalizes across inputs to enable the pipelined hardware acceleration. The importance score captures the **structural sparsity** of the model, identifying channels that are consistently critical. In contrast, the quantization error may not have this per-channel feature.
>
> **For weights**, our weight selection metric $I = W^2 / (H^{-1})^2$ is derived from GPTQ. It explicitly models the sensitivity of the global loss function to perturbation, which theoretically supersedes local L2 quantization error minimization.
>
> We further clarify that the SQ-format is not a specific algorithm, but rather a data format that supports multiple algorithms, benefiting from hardware-software co-design. We agree that error can serve as a potentially new quantization algorithm, and once it possesses the features that benefit from SQ-format, we can support its implementation.
>
> We will highlight this point in the revised paper to reflect the future extensibility of SQ-format.
>
> > **Question 5: Why the static activation strategy prefers smaller bank sizes?**
> >
>
> We thank you for this insightful observation based on our empirical results. We attribute this preference to **the need to compensate for the lack of runtime dynamism**.
>
> As hypothesized, the static strategy relies on a fixed mask derived from calibration, losing the ability to adapt to input-specific outliers at runtime. A smaller bank_size acts as a compensatory mechanism by providing finer quantization granularity. With smaller banks, the system calculates quantization parameters for smaller groups of channels, allowing it to fit local activation distributions more tightly than larger blocks.

---

### Official Review · Reviewer_3cJk · 2025-10-29

**Soundness:** 2
**Presentation:** 3
**Contribution:** 3
**Rating:** 4
**Confidence:** 4

**Summary:**

Post-training quantization (PTQ) is crucial for LLM deployment, but current hardware makes low-bit quantization and sparsification hard to balance for accuracy and efficiency (e.g., W4A8 offers similar peak TOPS as W8A8; GPU 2:4 sparsity often hurts accuracy). The authors propose a unified Sparse‑Quantized Format (**SQ‑format**) that leverages high‑precision sparse acceleration and extends it to low‑precision matmul, enabling static compression for outlier‑skewed activations and yielding Pareto gains in performance vs. throughput on existing GPUs and future hardware. They report state‑of‑the‑art PTQ results, specify required hardware support, and provide design insights for next‑generation AI accelerators.

**Strengths:**

1. This paper is well organized.
2. This paper proposes a novel quantization format.
3. The proposed method shows SOTA performance.

**Weaknesses:**

1. I think the results of this paper are not easy to reproduce, since the authors do not include code. It is hard to believe the training-free approach can achieve much better performance than training-based SpinQuant, as demonstrated in the paper.
2. The baseline performance of this paper is inconsistent with their original paper.
3. The authors do not include E2E speedup results and memory costs of the proposed method. This is very important for the application of the proposed format. Only theatrical analysis is not reasonable.
4. This paper has claimed that the proposed method supports FP quantization. I believe they should include results for such a setting.

**Questions:**

N/A

---

> ### Author Response · Authors · 2025-11-22
>
> Thank you for your feedback and for acknowledging the strengths of our paper! We have summarized the weaknesses and questions and respond as follows:
>
> > **Weakness 1 & 2: Reproducibility & comparison with SpinQuant.**
> >
>
> We thank you for your comment regarding reproducibility. We commit to releasing our full PTQ framework upon acceptance to ensure the results can be verified.
>
> To ensure a fair comparison, in this paper, we re-evaluate all baselines using their official open-source repositories within a unified environment, rather than simply copying numbers from their original papers. We respectfully present the comparison of our SpinQuant reproduction against the original paper's reported values below to validate our numbers:
>
> **Rebuttal Table 1: Accuracy (%) comparison of SpinQuant W4A8 on Llama-3-8B.**
>
> | **Model** | **Method** | **ARC-e** | **ARC-c** | **PIQA** | **HellaSwag** | **WinoGrande** | **OBQA** |
> | --- | --- | --- | --- | --- | --- | --- | --- |
> | Llama-3-8B | SpinQuant (Our reproduction) | 75.08 | 51.78 | 80.20 | 77.71 | 74.19 | **44.4** |
> |  | SpinQuant (Original paper) | 76.5 | 54.0 | 79.6 | 78.1 | 72.4 | **56.4** |
>
> As shown, the results match closely across most datasets. A significant discrepancy is observed only on OBQA, an inconsistency that has also been raised in the issues of their official repository. We hypothesize this stems from a metric mismatch: the original paper likely reports acc for this task, whereas we consistently used acc_norm across all evaluations.
>
> We understand your surprise that a training-free method outperforms a training-based one. SpinQuant uses rotation matrices to smooth outliers. However, rotation does not eliminate outlier energy; it merely spreads it across the channel. For extreme outliers, this spreading can expand the quantization range for all elements, effectively raising the noise floor for the entire vector. In contrast, SQ-format adopts an isolation strategy. By keeping outliers in exact INT8 and quantizing the rest in INT4, we prevent the outliers from distorting the quantization range of the dense part.
>
> > **Weakness 3: No end-to-end acceleration & memory footprint.**
> >
>
> We thank you for raising concerns regarding end-to-end performance and memory costs. We address these aspects below with real-world profiling data.
>
> 1. **End-to-End Latency & Bandwidth Reduction**
>
> We profile our static activation CUDA implementation on GPUs using the WikiText dataset to measure the real-world impact of SQ-format on memory bandwidth utilization and end-to-end latency. The results are in Rebuttal Table 2.
>
> **Rebuttal Table 2: End-to-end prefilling latency & effective bandwidth on GPUs.**
>
> | **Model** | **Type** | **Sparsity** | **Time** | **Effective BW (GB/s)** | **Speedup vs W4A8** |
> | --- | --- | --- | --- | --- | --- |
> | Llama-3-8B | W4A8 | - | 48s | 34.69 | 1x |
> |  | SQ-format | 0.75 | 42s | 36.90 | 1.14x |
> |  | SQ-format | 0.875 | 41s | 38.19 | 1.17x |
> |  | W4A4 | - | 38s | 40.61 | 1.26x |
> | Llama-3-70B | W4A8 | - | 10min8s | 10.84 | 1x |
> |  | SQ-format | 0.75 | 6min29s | 16.94 | 1.56x |
> |  | SQ-format | 0.875 | 5min55s | 18.54 | 1.71x |
> |  | W4A4 | - | 5min16s | 20.86 | 1.92x |
>
> The results show SQ-format significantly reduces inference latency compared to the W4A8 baseline. For Llama-3-70B, SQ-format $(s=0.875)$ achieves a 1.71x speedup, approaching W4A4. For Llama-3-8B, we observe a consistent 1.17x speedup.
>
> 2. **Memory Footprint Analysis**
>
> We clarify that the extra memory footprint of the masks is negligible for the static activation strategy because the mask is defined per-channel (1-bit per channel) rather than per-element.
>
> To demonstrate scalability, we calculate the exact storage size of the static masks across different model scales (7B to 70B) in Rebuttal Table 3. This confirms that maintaining these masks imposes no significant burden on memory, especially considering the bandwidth benefits for weights and computational benefits for activations.
>
> **Rebuttal Table 3: Static Mask Storage Overhead.**
>
> | **Model** | **Setting** | **Static Mask Size (MB)** |
> | --- | --- | --- |
> | Llama-2-7B | W4A(SQ-0.75) | 1.09 |
> | Llama-3-8B | W4A(SQ-0.75) | 1.19 |
> | Qwen3-30B | W4A(SQ-0.75) | 28.97 |
> | Llama-3-70B | W4A(SQ-0.75) | 5.94 |
>
> We will include these key data in the revised paper to support the theoretical benefits of the SQ-format.

---

> ### Author Response · Authors · 2025-11-22
>
> > **Weakness 4: No FP results while claimed.**
> >
>
> We thank you for pointing out the need for empirical results on FP quantization. We confirm that SQ-format is a general structural format supporting both INT and FP data types. Our initial experimental focus is primarily on INT because we specifically aim to solve the critical hardware-algorithm gap where schemes like W4A8 must be emulated using W8A8 data paths.
>
> During the rebuttal phase, we have run preliminary experiments applying SQ-format to the weights of the DeepSeek-R1 (685B), which utilizes an FP8 base format. These results validate the applicability of SQ-format in the FP domain. We apply weight SQ-format with high-precision FP8, low-precision FP4, bank_size=64 and 8x sparsity ($s=0.875$), resulting in a W(SQ5)A8 config. The accuracy results are provided in Rebuttal Table 4.
>
> **Rebuttal Table 4: Accuracy (%,** $\uparrow$**) and PPL (**$\downarrow$**) on DeepSeek-R1, setup: W(SQ5)A8.**
>
> | setup / benchmark | GPQA_D | Math-500 | GSM8K (5shot) | ARC_e | ARC_c | HellaS. | PIQA | OBQA | Wino. | AGIEval | WikiText2 (PPL) | Lambada (PPL) |
> | --- | --- | --- | --- | --- | --- | --- | --- | --- | --- | --- | --- | --- |
> | DeepSeek-R1 (baseline) | 73.637 | 97.4 | 95.83 | 85.52 | 64.42 | 87.43 | 84.98 | 48.4 | 79.95 | 70.22 | 3.33 | 12.67 |
> | SQ-format (W(SQ5)A8) | 73.485 | 97.2 | 96.21 | 85.52 | 63.99 | 87.19 | 85.31 | 46.6 | 79.45 | 69.58 | 3.39 | 12.77 |
>
> Due to resource constraints, we could not provide a full ablation study like Table 1 at this stage. We commit to continuing this exploration in the rebuttal phase, and the revised paper will include more FP experimental results and analysis.

---

> > ### Comment · Reviewer_3cJk · 2025-11-24
> >
> > Thanks for the reply.
> > * I believe it's better to align the settings with baseline papers and copy their evaluation results directly.
> > * The E2E speedup is *too weak*, and no detailed settings.
> >
> > Considering the above concerns and the contribution of this work, I think the current score is appropriate.

---

> > > ### Author Response · Authors · 2025-11-25
> > >
> > > Thank you for your continued engagement with our paper!
> > >
> > > 1. **Baseline methodology**
> > >
> > > We agree that referencing original reported values is valuable where there is overlap in models or datasets. However, we opt for reproduction to ensure a consistent evaluation environment and to extend the comparison to generative benchmarks like GSM8k and AGIEval, which typically suffer more severe degradation after quantization and are often omitted in baseline papers.
> > >
> > > To address your concern, we commit to adding a discussion section in the Appendix of the revised paper. This section will tabulate the original reported values from baseline papers alongside our reproduced results, providing full transparency on where they align or diverge.
> > >
> > > 2. **Strength of E2E speedup**
> > >
> > > Regarding the E2E speedup, we apologize for not providing a detailed setup in our first response. We clarify that our reported latency is based on the static activation strategy, which is the most efficient instance of SQ-format for current GPUs, whereas others rely on the proposed dedicated hardware. The bank_size is 64 for all SQ-format experiments in Rebuttal Table 2.
> > >
> > > We respectfully argue that the speedup is significant when viewed against the W4A4 theoretical limit. As shown in Rebuttal Table 2, for Llama-3-70B, SQ-format achieves a 1.71x speedup, which captures ~89% of the maximum possible speedup offered by pure W4A4 (1.92x). For Llama-3-8B, an acceleration of nearly 92% of W4A4 has also been achieved. Crucially, we achieve this speed without the associated accuracy collapse. As shown in Rebuttal Table 3, SQ-format offers a superior trade-off.
> > >
> > > **Rebuttal Table 3: Pareto comparison of speedup vs. average accuracy.**
> > >
> > > | **Model** | **Method** | **Setting** | **Speedup (vs. W4A8)** | **Avg. Acc (Non-gen)** | **Avg. Acc (Gen)** |
> > > | --- | --- | --- | --- | --- | --- |
> > > | **Llama-3-8B** | W4A8 | GPTQ | 1x | 67.86% | 36.87% |
> > > |  | SQ-format | $s=0.5$ | 1.07x | 67.69% (-0.17%) | 36.31% (-0.56%) |
> > > |  | SQ-format | $s=0.75$ | 1.14x | 67.48% (-0.38%) | 36.64% (-0.23%) |
> > > |  | SQ-format | $s=0.875$ | 1.17x | 66.69% (-1.17%) | 34.93% (-1.94%) |
> > > |  | W4A4 | GPTQ | 1.26x | 64.23% (-3.63%) | 29.52% (-7.35%) |
> > > | **Llama-3-70B** | W4A8 | GPTQ | 1x | 73.86% | 61.66% |
> > > |  | SQ-format | $s=0.5$ | 1.31x | 73.84% (-0.02%) | 61.35% (-0.31%) |
> > > |  | SQ-format | $s=0.75$ | 1.56x | 73.76% (-0.10%) | 61.08% (-0.58%) |
> > > |  | SQ-format | $s=0.875$ | 1.71x | 72.85% (-1.01%) | 60.69% (-0.97%) |
> > > |  | W4A4 | GPTQ | 1.92x | 71.93% (-1.93%) | 57.71% (-3.95%) |
> > >
> > > We will include these data in the revised paper to better demonstrate the effects of Pareto improvement. We hope that our response and the new experimental results have adequately addressed your concerns. We would greatly appreciate it if you could consider re-evaluating our work.

---

### Official Review · Reviewer_Bt9D · 2025-10-30

**Soundness:** 3
**Presentation:** 3
**Contribution:** 3
**Rating:** 6
**Confidence:** 4

**Summary:**

This paper addresses the challenge of efficiently deploying large language models under both quantization and sparsity constraints, aiming to reduce model size and computation while maintaining accuracy. The authors propose SQ‑format, a unified sparse-quantized data format that encodes weights and activations in mixed precision and uses masks to indicate high-precision elements, allowing block-wise efficient computation. They introduce algorithms to determine which elements to quantize at low precision for weights and activations, using either static or dynamic mask strategies. Extensive experiments on LLaMA‑3 and Qwen‑3 models show that SQ‑format achieves comparable or better accuracy than prior quantization or sparsity methods while improving throughput and hardware efficiency. SQ‑format provides a hardware-friendly approach for mixed sparse-quantized LLMs, enabling practical deployment on accelerators without sacrificing performance.

**Strengths:**

- SQ‑format combines sparsity and quantization in a single representation, facilitating efficient computation on modern hardware.

- The paper carefully considers practical deployment, proposing both static and dynamic mask strategies to balance accuracy and efficiency.

- Evaluations on multiple LLMs (8B–70B) with standard benchmarks demonstrate that SQ‑format preserves accuracy while improving throughput.

- Addresses a key deployment challenge for LLMs, bridging the gap between algorithmic innovations and hardware execution.

**Weaknesses:**

- Limited comparison to extreme low-bit settings: The paper mainly evaluates INT4–INT8 and moderate sparsity; performance in ultra-low bit scenarios (e.g., W4A4) is unclear.

- Complexity of mask design: Dynamic mask selection may introduce runtime overhead, and static masks require careful calibration, which may complicate practical adoption.

- Specific to current hardware: While hardware-friendly, the proposed format is tuned to modern GPUs; applicability to other accelerators (TPU, AI chips) is not fully validated.

- Additional storage overhead: Maintaining masks for sparse/high-precision elements increases memory usage, which may be non-trivial for very large models.

- The algorithm design is simple and lacks novelty, but it imposes a very heavy burden on deployment. Although the final results are indeed good, there are still concerns about the future prospects of this method.

**Questions:**

Please refer to the weaknesses above.

---

> ### Author Response · Authors · 2025-11-22
>
> Thank you for your feedback and for acknowledging the strengths of our paper! We have summarized the weaknesses and questions and respond as follows:
>
> > **Weakness 1: Limited comparison with extremely low bit settings.**
> >
>
> We thank you for this critical comment. We apologize if the dispersed presentation of low-bit results caused confusion. Our paper does indeed cover ultra-low bit settings, and we have consolidated these results below to demonstrate that SQ-format significantly outperforms standard W4A4 at equivalent or lower bit-widths.
>
> - Figures 6 & 7 include the average accuracy results of Llama-3-8B when low precision being INT2 and INT3.
> - Table 3 in Appendix A includes detailed results for Llama-2-7B with low precision of INT2 and INT3. These results are compared with W4A8. In cases of > 4x sparsity, its equivalent bit width is lower than that of W4A4.
>
> At a low precision of INT3, accuracy can still be maintained under dynamic activation SQ-format at 4x or 8x sparsity. However, when faced with even lower precision (such as INT2, especially high INT4 low INT2), the model's performance declines significantly. Therefore, this paper takes W4A4 as a key performance-accuracy Pareto turning point for comparison.
>
> > **Weakness 2: Complexity of mask design.**
> >
>
> We thank you for highlighting the practical challenges of mask implementation. We address your concerns regarding runtime overhead and calibration sensitivity below.
>
> 1. **Dynamic Mask**
>
> We agree that dynamic mask selection introduces overhead on current GPUs due to the lack of native support. However, the SQ-format is co-designed for next-generation accelerators where this process is pipelined using dedicated hardware (Figure 4).
>
> To demonstrate that this hardware support does not introduce excessive area cost, we implement the proposed SQ-MAC unit in RTL and synthesized it using the TSMC 12nm process library. We compared it against a standard integer MAC array under an ISO-I/O-bandwidth constraint (12-bit I/O). The baseline is standard INT6 MAC Array (6-bit weight + 6-bit activation). We implement SQ-format MAC with 4x Sparse weight + INT8 activation, effectively matching 12-bit I/O bandwidth.
>
> **Rebuttal Table 1: Area Synthesis Comparison (Normalized by Adder Area).**
>
> | **Area \ Format** | **SQ-format (Ours)** | **Standard INT6 MAC** |
> | --- | --- | --- |
> | Multiplier | 0.42 | 2.23 |
> | Adder | 0.24 | 1 |
> | Accumulator | 0.21 | 0.002 |
> | Low bit mult. | 0.42 | -- |
> | Gather Unit | 0.783 | -- |
> | Total Area | 2.073 | 3.232 |
>
> As shown in Rebuttal Table 1, even accounting for the gather unit, which handles the dynamic/sparse mask logic, the total area is 35.8% smaller than the standard baseline. This demonstrates that the complexity of the mask design translates to efficient, low-area hardware logic, not a deployment burden.
>
> 2. **Static Mask**
>
> Regarding the static strategy, we respectfully clarify that it does not require careful calibration. The static mask relies on channel-wise outlier statistics, which are structural features of the model and easy to stably capture. As shown in Table 7 of Appendix A, we evaluate W4A(SQ6) performance on Llama-3-8B while varying the calibration set size from 8 to 512 samples. The benchmark performance fluctuations are negligible (e.g., GSM8k accuracy remains stable around 39-41%). This confirms that the activation outlier distribution is easily modeled with limited data, making the calibration process robust and simple for practical adoption.

---

> ### Author Response · Authors · 2025-11-22
>
> > **Weakness 3: No applicability to Non-GPU accelerators (TPU, AI Chips).**
> >
>
> We thank you for your forward-looking comment. While we utilize GPUs for prototyping due to their dominance in current LLM deployment, SQ-format addresses a fundamental challenge in the hardware landscape: the fragmentation of sparse representations. Its design is highly applicable to diverse AI accelerators beyond GPUs.
>
> 1. **Bridging the Sparse Format Gap**
>
> Currently, there is no consensus on hardware-level sparse formats, which limits the portability of sparse models. Different chips utilize highly divergent representations. For example, [1] uses CSR (Compressed Sparse Row) formats; [2] uses CSC (Compressed Sparse Column coding) formats; [3,4] use formats specialized for their own dataflows.
>
> SQ-format acts as a unified representation. Because it enforces structured regularity via fixed bank_size and sparsity, it can be efficiently mapped to both CSR and CSC hardware backends. This regularity simplifies the control logic required by chips compared to processing totally unstructured sparsity.
>
> 2. **Applicability to TPUs**
>
> Standard TPUs are optimized for dense computations and do not natively support general sparse matrix multiplication, focusing instead on specific ops like sparse embeddings.
>
> **References**
>
> [1] SambaNova SN10 RDU: Accelerating Software 2.0 with Dataflow, Hotchips 2021.
>
> [2] Chen, Y. H., et al. "v2: A flexible accelerator for emerging deep neural networks on mobile devices., 2019, 9." DOI: https://doi.org/10.1109/JETCAS (2019): 292-308.
>
> [3] Cerebras Architecture Deep Dive: First Look Inside the HW/SW Co-Design for Deep Learning, Hotchips 2022.
>
> [4] Tenstorrent's Holistic Stack Of AI Innovation, https://moorinsightsstrategy.com/wp-content/uploads/2020/10/Tenstorrents-Holistic-Stack-Of-AI-Innovation-By-Moor-Insights-And-Strategy.pdf.
>
> > **Weakness 4: Storage overhead of masks for larger models.**
> >
>
> We thank you for raising the concern regarding storage overhead. We clarify that the memory footprint of the masks is negligible for the static activation strategy, because the mask is defined per-channel (1-bit per channel) rather than per-element.
>
> To demonstrate scalability, we calculate the exact storage size of the static masks across different model scales (7B to 70B) shown in Rebuttal Table 2. This confirms that maintaining these masks imposes no significant burden on memory, especially considering the bandwidth benefits for weights and computational benefits for activations.
>
> **Rebuttal Table 2: Static Mask Storage Overhead.**
>
> | **Model** | **Setting** | **Static Mask Size (MB)** |
> | --- | --- | --- |
> | Llama-2-7B | W4A(SQ-0.75) | 1.09 |
> | Llama-3-8B | W4A(SQ-0.75) | 1.19 |
> | Qwen3-30B | W4A(SQ-0.75) | 28.97 |
> | Llama-3-70B | W4A(SQ-0.75) | 5.94 |
>
> > **Weakness 5: Algorithm simplicity & deployment burden**
> >
>
> We respectfully disagree with the characterization of the method as imposing a heavy burden. On the contrary, as a hardware-software collaborative design, we argue that algorithmic simplicity is a fundamental virtue for hardware deployment, particularly for next-generation AI accelerators.
>
> **Complex quantization or pruning algorithms often require sophisticated control flow** (e.g., if-else branching for dynamic execution) or complex memory management. In hardware design, such complexity translates to Warp Divergence on GPUs or Pipeline Bubbles on systolic arrays, severely degrading real-world throughput despite theoretical FLOP reductions. Our design, specifically the **fixed bank_size and sparsity allows it to map directly to basic hardware primitives** (MUX and MAC). It requires no complex branch prediction or dynamic scheduling logic.
>
> The burden of deployment is minimized because the complexity is handled offline. The identification of high-precision elements happens during the calibration phase (which takes only minutes).

---

### Official Review · Reviewer_eKea · 2025-11-01

**Soundness:** 3
**Presentation:** 2
**Contribution:** 3
**Rating:** 4
**Confidence:** 3

**Summary:**

The paper introduces SQ‑format, a unified, hardware‑friendly data format that combines sparsification and quantization for large language models (LLMs). It encodes a tensor as a sparse high‑precision component for critical values and a dense low‑precision component for the rest, thereby achieving a better trade‑off between accuracy and throughput. The authors present algorithms to apply SQ‑format to weights and activations, and provide hardware design insights. Empirical results on multiple LLMs show that SQ‑format reaches near W4A8 accuracy while maintaining W4A4‑level throughput.

**Strengths:**

This paper introduces the SQ-format, which integrates sparsification and quantization into a unified data representation, bridging the gap between algorithmic compression and hardware efficiency for improved throughput and accuracy.

This paper introduces SQ-format for weights and activations, achieving significant throughput gains while maintaining near W4A8-level accuracy, effectively balancing efficiency and performance.

This paper introduces a static activation splitting strategy that reduces runtime overhead, making SQ-format more practical for deployment on current AI accelerators.

**Weaknesses:**

1. The paper proposes a unified sparse + quantized format (SQ‑format) combining high‑precision for critical values and low‑precision + sparsity for the rest. However, previous work, such as SpQR: A Sparse‑Quantized Representation for Near‑Lossless LLM Weight Compression (Dettmers et al., 2023), already investigates the idea of preserving a small subset of weights in high precision and quantizing the remainder. While SQ‑format adds the “bank” structure and hardware‑mapping discussion, the paper could do more to clearly highlight what is novel beyond those prior methods.

2. The authors argue that SQ‑format is “hardware‑friendly” and outline required hardware support, but they offer only simulation or theoretical throughput estimates—not measured latency, power, or memory‑bandwidth results on real GPUs or accelerators.

**Questions:**

1. How do you select the ratio of high‑precision elements in the “bank” structure (bank size  b, sparsity rate s) as model size increases (e.g., 8B → 70B)?

2. Can you quantify the extra memory bandwidth or branching overhead introduced by decoding the SQ‑format (high/low precision mix + sparsity mask) compared to a uniform low‑precision format?

3. If activation distributions shift (e.g., instruction‑tuned or domain‑adapted LLMs), how robust is the static activation split strategy, and what is the accuracy or latency impact?

---

> ### Author Response · Authors · 2025-11-22
>
> Thank you for your feedback and for acknowledging the strengths of our paper! We have summarized the weaknesses and questions and respond as follows:
>
> > **Weakness 1: Similar work already exists, and the novelty is not highlighted.**
> >
>
> Thanks for your comment! While we acknowledge SpQR also adopts the concept of **outlier isolation**, SQ-format differs fundamentally in data layout and hardware execution paradigm, addressing the key efficiency bottlenecks of SpQR:
>
> **Structured Representation:** SpQR relies on unstructured data representation for outliers. This necessitates storing explicit 16-bit column indices for every outlier, causing irregular memory access patterns that are inefficient on GPUs. In contrast, SQ-format utilizes a bank-based sparsity structure. This structural constraint enforces aligned memory access, eliminating the need for heavy index metadata and enabling effective memory transactions, which is critical for actual hardware speedup.
>
> **Hardware Determinism:** SpQR requires complex runtime load-balancing algorithms to handle irregular sparsity. SQ-format's banked structure ensures deterministic workload distribution. This hardware-friendliness is not just a minor addition; it transforms the method from a compression-oriented technique (SpQR) to a throughput-oriented data format compatible with next-generation hardware primitives.
>
> We will highlight this point in the revised paper, emphasizing the contributions at different levels compared to SpQR.

---

> ### Author Response · Authors · 2025-11-22
>
> > **Weakness 2 & Question 2: Only simulated performance provided.**
> >
>
> Thanks for your comment! Since the dedicated hardware primitives proposed in Section 2.3 do not yet exist in commercial GPUs, measuring end-to-end wall-clock latency for SQ-format is not physically representative of the proposed architecture's potential. To move beyond theoretical estimates and substantiate hardware-friendliness, we conduct three experiments to demonstrate the advantages of SQ-format.
>
> 1. **Hardware Synthesis**
>
> To address feasibility and power concerns, we implement the proposed SQ-MAC unit in RTL and synthesized it using the TSMC 12nm process library. We compare it against a standard integer MAC array under an ISO-I/O-bandwidth constraint (12-bit I/O). The baseline is standard INT6 MAC Array (6-bit weight + 6-bit activation). We implement SQ-format MAC with 4x Sparse weight + INT8 activation, effectively matching 12-bit I/O bandwidth.
>
> **Rebuttal Table 1: Area Synthesis Comparison (Normalized by Adder Area).**
>
> | **Area \ Format** | **SQ-format (Ours)** | **Standard INT6 MAC** |
> | --- | --- | --- |
> | Multiplier | 0.42 | 2.23 |
> | Adder | 0.24 | 1 |
> | Accumulator | 0.21 | 0.002 |
> | Low bit mult. | 0.42 | -- |
> | Gather Unit | 0.783 | -- |
> | Total Area | 2.073 | 3.232 |
>
> As shown in Rebuttal Table 1, SQ-format architecture achieves a 35.8% reduction in total silicon area. While SQ-format introduces extra logic for sparse gathering, this is significantly outweighed by the reduction in multiplier size. In hardware design, area reduction is a strong proxy for static and dynamic power savings, confirming the design is physically efficient.
>
> 2. **Latency & Bandwidth Reduction**
>
> We profile our static activation CUDA implementation on GPUs using the WikiText dataset to measure the real-world impact of SQ-format on memory bandwidth utilization and end-to-end latency. The results are in Rebuttal Table 2.
>
> **Rebuttal Table 2: End-to-end prefilling latency & effective bandwidth on GPUs.**
>
> | **Model** | **Type** | **Sparsity** | **Time** | **Effective BW (GB/s)** | **Speedup vs W4A8** |
> | --- | --- | --- | --- | --- | --- |
> | Llama-3-8B | W4A8 | - | 48s | 34.69 | 1x |
> |  | SQ-format | 0.75 | 42s | 36.90 | 1.14x |
> |  | SQ-format | 0.875 | 41s | 38.19 | 1.17x |
> |  | W4A4 | - | 38s | 40.61 | 1.26x |
> | Llama-3-70B | W4A8 | - | 10min8s | 10.84 | 1x |
> |  | SQ-format | 0.75 | 6min29s | 16.94 | 1.56x |
> |  | SQ-format | 0.875 | 5min55s | 18.54 | 1.71x |
> |  | W4A4 | - | 5min16s | 20.86 | 1.92x |
>
> The results show SQ-format significantly reduces inference latency compared to the W4A8 baseline. For Llama-3-70B, SQ-format $(s=0.875)$ achieves a 1.71x speedup, approaching W4A4. For Llama-3-8B, we observe a consistent 1.17x speedup.
>
> 3. **Extra Gathering Operations Profiling**
>
> In this paper, since the static activation-based SQ-format is simulated using GPUs, we need the gather operator to separate high and low precision data according to the mask without introducing dedicated hardware. The results are demonstrated in Rebuttal Table 3.
>
> **Rebuttal Table 3: Latency breakdown of single SQ-format matmul operation.**
>
> | **Model** | **Phase** | **Time (ms)** |
> | --- | --- | --- |
> | Llama-3-8B | Gathering | 0.5806 |
> |  | Activation quantization | 0.4209 |
> |  | Matmul | 14.4261 |
>
> The results show that the overhead introduced by the gathering phase is negligible. While our proposed hardware (Section 2.3) would further hide this latency, these results confirm that the SQ-format is inherently efficient.
>
> We will include these key data in the revised paper to support the theoretical benefits of the SQ-format.

---

> ### Author Response · Authors · 2025-11-22
>
> > **Question 1: Bank & sparsity decision among different model sizes.**
> >
>
> We thank you for your insightful question regarding scalability. We agree that as a hardware-software collaborative data format, the choice of bank_size and sparsity in SQ-format is crucial for generalization across models.
>
> 1. **Choices for bank_size**
>
> Currently, our choices for bank_size are primarily based on the practical performance testing of models of different scales. As model size increases from 8B to 70B, we do not observe a need to adjust the bank_size. The acceptable configuration remains consistent. The distribution of outliers in LLMs tends to be structurally heavy-tailed rather than scale-dependent. A bank size of 64 is sufficiently large to smooth out local variations in sparsity while being small enough to capture clustered outliers (discussed in Section 4.1).
>
> To demonstrate the generalizability of bank_size = 64 on larger models, we provide the benchmarking results of running W(SQ5)A8 on DeepSeek-R1 (685B) in Rebuttal Table 4, with high-precision FP8, low-precision FP4, and 8x sparsity ($s=0.875$). The accuracy results support that the choice of bank_size = 64 is relatively independent of the model size.
>
> **Rebuttal Table 4: Accuracy (%,** $\uparrow$**) and PPL (**$\downarrow$**) on DeepSeek-R1, setup: W(SQ5)A8.**
>
> | setup / benchmark | GPQA_D | Math-500 | GSM8K (5shot) | ARC_e | ARC_c | HellaS. | PIQA | OBQA | Wino. | AGIEval | WikiText2 (PPL) | Lambada (PPL) |
> | --- | --- | --- | --- | --- | --- | --- | --- | --- | --- | --- | --- | --- |
> | DeepSeek-R1 (baseline) | 73.637 | 97.4 | 95.83 | 85.52 | 64.42 | 87.43 | 84.98 | 48.4 | 79.95 | 70.22 | 3.33 | 12.67 |
> | SQ-format (W(SQ5)A8) | 73.485 | 97.2 | 96.21 | 85.52 | 63.99 | 87.19 | 85.31 | 46.6 | 79.45 | 69.58 | 3.39 | 12.77 |
>
> 2. **Choices for sparsity**
>
> Sparsity is primarily determined by the target hardware throughput rather than model sensitivity. As discussed in Section 4.2, to achieve a pareto improvement over W8A8, the sparsity must be at least $0.75$ (4x sparse) to mask the latency of the high-precision path. Our experiments in Table 1 and Figure 6 show that even at $s=0.9375$ (16x sparse), SQ-format can maintain high upon appropriate bank_size.
>
> This stability is pivotal for our hardware design. We have adopted bank_size = 64 as a preferred architectural parameter for our next-generation AI accelerator. This eliminates the need for complex dynamic tiling logic, simplifying the hardware control path significantly.
>
> > **Question 3: Effect when activation distributions shift.**
> >
>
> We thank the reviewer for raising the important scenario of domain shift. SQ-format is a data format specialized for PTQ. For models with significant distribution shift, the standard procedure is to re-generate the static mask using a calibration set within the target domain.
>
> The feasibility of re-calibration relies on its speed. Unlike optimization-heavy PTQ methods, the computation for SQ-format is algorithmically lightweight. As described in Algorithm 2, the mask generation process is non-iterative. It requires only a single forward pass over the calibration set to accumulate channel statistics, followed by a simple Top-K calculation to determine the mask.
>
> To quantify this, we measure the wall-clock time for the full process on the same hardware and calibration setup in Rebuttal Table 5.
>
> **Rebuttal Table 5: PTQ Calibration Time.**
>
> | **Model** | **SQ-format (Ours)** | **SpinQuant** |
> | --- | --- | --- |
> | Llama-3-8B | ~12 mins | ~40 mins |
> | Llama-3-70B | ~48 mins | ~4 hrs |
>
> From the results in Rebuttal Table 5, SQ-format is roughly 4x faster than SpinQuant. This makes re-calibration a negligible engineering cost in deployment pipelines.

---

### Author Response · Authors · 2025-11-28
**Global Response: Revised Paper Uploaded**

Dear reviewers,

We sincerely thank you for your constructive comments and the time invested in reviewing our work. We have uploaded the revised paper. We hope the additional experiments, particularly the real-world end-to-end latency which supports the accuracy-speed Pareto improvement (Figure 5), can address your concerns regarding hardware realism.

Furthermore, we would like to highlight our novelty in hardware-software co-design. By validating the design through RTL synthesis and extensive config exploration on model performance, we demonstrate that SQ-format significantly enhances the physical implementability of structured hybrid-precision PTQ, distinguishing our work from previous unstructured methods.

We respectfully hope that our response and revision have successfully addressed your concerns. We look forward to your feedback and are happy to engage in any further discussion.

Best regards,
Paper2231 Authors

---

### Author Response · Authors · 2025-12-03
**Summary of Our Paper and Discussion**

We sincerely thank all reviewers for their constructive feedback, and the ACs, SACs, and PCs for their thoughtful consideration of our submission.

**Summary of our contribution:**

- We identify the hardware-algorithm gap where current hybrid-precision schemes like W4A8 fail to deliver theoretical throughput gains on existing hardware.
- We propose SQ-format, a unified sparse-quantized representation that decomposes tensors into sparse high-precision and dense low-precision components.
- We demonstrate that SQ-format achieves a Pareto improvement, bridging the gap between W4A8 accuracy and W4A4 throughput.
- We provide a complete hardware-software co-design, including a static activation strategy and a dedicated hardware unit architecture, to ensure physical realizability.

We are grateful that reviewers recognized the **novelty of integrating sparsification and quantization into a unified format** (eKea, 3cJk), the **practicality of addressing key deployment challenges** (Bt9D), the **SOTA performance** **achieved** (3cJk), and the **well-written presentation** **and extensive experiments** (CDnV).

**How we address the reviewer’s concerns:**

We have made every effort to address all points raised with new experiments and data:

- **Physical feasibility beyond simulation (eKea, CDnV):** We conducted RTL synthesis using the TSMC 12nm process library. The results demonstrate that our dedicated SQ-MAC unit achieves a 35.8% area reduction compared to a standard INT6 MAC array under ISO-I/O-bandwidth constraints. This serves as strong evidence that SQ-format is a physically efficient design, not just a theoretical promise.
- **Real-world End-to-End Speedup (3cJk, CDnV):** We profiled the end-to-end latency of our static activation strategy on GPUs. The results show a 1.71x speedup for Llama-3-70B, capturing approximately 89% of the theoretical speedup limit of W4A4. This validates our claim of Pareto superiority: achieving near-W4A4 speed with significantly higher accuracy.
- **Comparison with Baselines and Reproducibility (3cJk, CDnV):** We addressed the concern regarding SpinQuant by reproducing their results and identifying a metric mismatch (acc vs. acc_norm) in their original reporting. We confirmed that our training-free SQ-format outperforms training-based methods by effectively isolating outliers rather than smoothing them.
- **Generalization to FP and Large Models (Bt9D, 3cJk):** We extended our evaluation to DeepSeek-R1 (685B) using an FP8/FP4 hybrid-precision configuration. The results confirm that SQ-format is scalable to massive models and applicable to floating-point formats, not limited to integer quantization.
- **Storage and Complexity (Bt9D):** We clarified that the static mask storage overhead is negligible (e.g., only ~6MB for Llama-3-70B) due to its per-channel nature, and that the bank-based regularity ensures hardware determinism unlike unstructured approaches.

In light of these efforts, we are confident that the concerns regarding hardware realism and performance validation have been properly addressed.

We believe our contributions provide substantial value to the community by offering a blueprint for next-generation AI accelerators that natively support structured hybrid-precision. We respectfully ask that these points be taken into consideration, and we have incorporated all new data and suggestions into the revised paper.

Thank you again for your time and consideration.

Regards,

Paper2231 Authors

---

### Meta-Review · Area_Chair_tY5K · 2026-01-06

**Summary:**

This submission proposes SQ-format, a unified sparse-quantized data format bridging the hardware-algorithm gap in LLM post-training quantization. By decomposing tensors into sparse high-precision and dense low-precision components, it leverages structured regularity to achieve Pareto improvement between accuracy and throughput, solving the longstanding trade-off in low-bit quantization and sparsification.

Reviewers broadly recognized the work’s practical impact, but also raised the concern about its novelty and the limited comparison of existing methods.

**Reviewer Concerns:**

The main concerns raised by reviewers centered on novelty justification, hardware realism, performance validation, and practical extensibility. In particular, some reviewers initially questioned whether SQ-format sufficiently differentiates itself from prior works (e.g., SpQR, QUIK) that also adopt outlier isolation, given the shared high-level concept. Related concerns included the reliance on simulation for initial throughput estimates, the absence of end-to-end speedup data, and uncertainty about applicability to non-GPU accelerators and floating-point formats.

Additional issues included mask storage overhead for large models, robustness to activation distribution shifts, and the lack of detailed ablation studies for floating-point quantization. Minor concerns involved baseline alignment with original papers and clarity on bank size selection rationale.

Based on the rebuttal and revised manuscript, some concerns were addressed. However, the novelty concern is still the major concern of this paper.

**Reviewer Scores:**

Based on the discussion and rebuttal, it is likely that reviewers with initial reservations would slightly adjusted their scores to reflect the comprehensive resolution of key concerns. Although the authors address some concerns in the rebuttal, the novelty concerns is not well addressed.

---

### Decision · Program_Chairs · 2026-01-26

Reject